# A Study of Capturing AMOC Regime Transition through Observation-Constrained Model Parameters

Zhao Liu[1,4], Shaoqing Zhang[1,2,3,4], Yang Shen[5], Yuping Guan[6,7], and Xiong Deng[8]

[1]Key Laboratory of Physical Oceanography, Ministry of Education/Institute for Advanced Ocean Study/Frontiers Science Center for Deep Ocean Multispheres and Earth System (DOMES), Ocean University of China, Qingdao, 266100, China
[2]Pilot National Laboratory for Marine Science and Technology (QNLM), Qingdao, 266237, China
[3]International Laboratory for High-Resolution Earth System Model and Prediction (iHESP), Qingdao, 266000, China
[4]College of Oceanic and Atmospheric Sciences, Ocean University of China, Qingdao, 266100, China
[5]College of Science, Liaoning University of Technology, Jinzhou, 121001, China
[6]State Key Laboratory of Tropical Oceanography, Chinese Academy of Sciences, Guangzhou, 510301, China
[7]College of Marine Sciences, University of Chinese Academy of Sciences, Beijing, 100049, China
[8]College of Intelligent Systems Science and Engineering, Harbin Engineering University, Harbin, 150001, China

*Correspondence to*: Shaoqing Zhang (szhang@ouc.edu.cn)

**Abstract.** The multiple equilibria are an outstanding characteristic of the Atlantic meridional overturning circulation (AMOC) that has important impacts on the Earth climate system appearing as regime transitions. The AMOC can be simulated in different models but the behavior deviates from the real world due to the existence of model errors. Here, we first combine a general AMOC model with an ensemble Kalman filter to form an ensemble coupled model data assimilation and parameter estimation (CDAPE) system, and derive the general methodology to capture the observed AMOC regime transitions through utilization of observational information. Then we apply this methodology designed within a "twin" experiment framework with a simple conceptual model that simulates the transition phenomenon of AMOC multiple equilibria, as well as a more physics-based MOC box model to reconstruct the "observed" AMOC multiple equilibria. The results show that the coupled model parameter estimation with observations can significantly mitigate the model deviations, thus capturing regime transitions of the AMOC. This simple model study serves as a guideline when a coupled general circulation model is used to incorporate observations to reconstruct the AMOC historical states and make multi-decadal climate predictions.

## 1 Introduction

The Atlantic meridional overturning circulation (AMOC), the core of the thermohaline circulation, is an essential component of the World Ocean circulations (e.g., Delworth and Greatbatch, 2000). One of its important characteristics is the existence of multiple equilibria (Mu et al., 2004). The research addressing this characteristic originates from Stommel (1961) who used two boxes with uniform temperature and salinity to simulate the equatorial ocean and the polar ocean respectively. This box model simulates multiple equilibria of thermohaline circulation including three steady solutions: a stable thermal mode, an unstable thermal mode (mainly driven by heat), and a stable haline mode (mainly controlled by salinity). Using an idealized box model has since become one of the most efficient approaches in the studies of AMOC simulations.

The idealized box model, although of limited applicability in simulating the entire Atlantic circulation or even the global circulation, provides the most basic explanation for some of the important characteristics of the AMOC (Scott et al., 1999). Besides Stommel's box model, which places two boxes side by side, Welander (1982) placed one box on top of the other to simulate the vertical structure of the real ocean. Then, the two-box model is extended to three boxes, and different three-box hemispheric models result in multiple equilibrium solutions (Birchfield, 1989; Guan and Huang, 2008; Shen et al., 2011). Also based on Stommel's box model, in some studies an additional box is added to simulate interhemispheric flows, constructing an idealized double-hemisphere model consisting of two high-latitude boxes and a low-latitude box. Multiple equilibria appear in such box models, and the transition between multiple equilibrium states is related to salt flux or freshwater flux (Rooth, 1982; Rahmstorf, 1996; Scott et al., 1999). Extending Rooth's box model, with the equatorial box and the polar box connected at depth, results in nine equilibrium solutions, four of which are stable (Welander, 1986). The double-hemisphere model is closer to the real AMOC than the hemispheric model regarding the cross-equatorial flow in the Atlantic and upwelling flows in the Southern Ocean.

The multiple equilibrium states of the AMOC have been confirmed not only in simple idealized box models but also in comprehensive ocean general circulation models (Fürst and Levermann, 2012). In addition to the four different equilibrium states obtained in ocean circulation models with two basins representing the idealized Atlantic and Pacific oceans (Marotzke and Willebrand, 1991), it is even more encouraging that multiple equilibria are first simulated in a complex ocean general circulation model (Bryan, 1986), followed by two steady states in a global model of the coupled ocean-atmosphere system (Manabe and Stouffer, 1988). While such a phenomenon of AMOC multiple equilibria as a reverse haline mode cannot be directly simulated in general circulation models (e.g., Stouffer et al., 2006; Weijer et al., 2019), it is instead replaced by a weak positive circulation or a collapsed AMOC state (e.g., Liu et al., 2013), generally referring to regime transitions.

Constrained by the limited measurement technique and time length, the direct observation of AMOC is in general scarce, in terms of its nature of rich spectrum especially addressing low-frequency (e.g., Delworth et al., 1993). The direct observation of AMOC is mainly from the RAPID-MOC/MOCHA (Meridional Overturning Circulation and Heatflux Array) mooring array, which has been conducted at 26° N since 2004 (Cunningham et al., 2007; Smeed et al., 2014). The scope of direct observation is difficult to cover the entire Atlantic Ocean, and it is difficult to achieve long-term continuous direct observation. Ocean temperature data could be used to derive a proxy index for the variability of the AMOC, so both observations from satellites and ocean temperature measurements from the ARGO program could be used to monitor AMOC, and historical variations of AMOC could be reconstructed from historical sea surface temperature (Zhang, 2008). Indicators representing AMOC can be established based on the physical relationship between AMOC and atmospheric indices or oceanic variables (e.g., Delworth et al., 2016; Caesar et al., 2018). Previous studies have compared and evaluated some of these indicators with direct observations of AMOC and the results indicate that this approach is feasible for AMOC reconstruction (Sun et al., 2020). However, the direct observations from RAPID or the ocean temperature measurements from the ARGO program are only available for about the last two decades, and the lack of multi-decadal observations makes it impossible to evaluate the multi-decadal AMOC variation. Besides, Paleoclimate records from marine sediments or ice cores are often used to investigate AMOC variations

(e.g., Rühlemann et al., 2004; Lynch-Stieglitz, 2017). Paleoclimate data can be used as observations of the AMOC on centurial and millennial timescales. Analyses of paleoclimate data reveal that the strength and pattern of AMOC changed between the glacial and interglacial periods (e.g., Bryan, 1986). The two equilibrium solutions in the work of Birchfield (1989) correspond to the modern ocean and the warm saline Cretaceous ocean, respectively. In summary, direct observations of AMOC are so scarce as to be unrepresentative in studies of multi-equilibria of AMOC at long time scales, and paleoclimate data have considerable uncertainty, so numerical simulations using ocean circulation models and coupled climate models are the main method to study the multiple equilibria of AMOC at present.

The transition between different equilibrium states is related to many factors, one of which is freshwater, the most commonly considered, starting with Stommel's box model that illustrates the effect of freshwater input on thermohaline circulation (Lambert et al., 2016). Changes in freshwater over a range of parameters may trigger shifts between different equilibrium states (e.g., Bryan, 1986; Marotzke and Willebrand, 1991; Nilsson and Walin, 2001; Stouffer et al., 2006; Nilsson and Walin, 2010). In addition to freshwater fluxes, the multiple equilibria may also be influenced by wind-driven gyre. The multiple equilibrium solutions in both Stommel's box model and Rooth's box model will be eliminated by a strong enough wind-driven ocean gyre (Longworth et al., 2005), and the same result can be obtained by replacing the buoyancy constraint with an energy constraint (Guan and Huang, 2008). AMOC transitions can occur due to external forcing or internal feedback (Klockmann et al., 2020). The external forcing applied in systems may include freshwater forcing (e.g., Cessi, 1994; Castellana et al., 2019), wind forcing (e.g., Ashkenazy and Tziperman, 2007; Kleppin et al., 2015), ice sheet forcing (e.g., Zhang et al., 2014; Mitsui and Crucifix, 2017), $CO_2$ forcing (e.g., Zhang et al., 2017). The physical processes in the model are changed by external forcing, resulting in the transition between different states of the AMOC. For the AMOC model without external forcing, the transition is triggered by complex internal interactions within the model, such as salt oscillations (Peltier and Vettoretti, 2014), internal oceanic processes (Sévellec and Fedorov, 2014), thermohaline oscillations (Brown and Galbraith, 2016), intermittencies in the sea-ice cover (Gottwald, 2021). Regardless of whether it is due to external forcing or internal feedback, AMOC transitions could be influenced by complex physical processes in models, and the parameters involved in these physical processes are usually fixed. However, due to an incomplete understanding of the physical processes and the error of the default parameter values, the numerical model is problematic in simulating AMOC multiple equilibria. This study addresses the problem that for long time scale AMOC reanalysis data, the AMOC multiple equilibrium states simulated by different models are different and do not fully represent the "real" AMOC multi-equilibrium transition path. How to simulate regime transition of AMOC with a model where influencing factors such as freshwater and wind-driven gyre change over time. Then the next key is how to make the simulation results closer to "reality" on the feature of regime transitions by constraining the parameter values with observation. Observation-constrained model parameters are no longer kept at fixed values but are constantly varying over time. The purpose of this paper is to explore whether the variations of observation-constrained parameters that allow the physical processes of model to evolve over time can bring the simulation results closer to the "observed" feature of regime transitions. The models in this paper are obtained by coupling AMOC box model with Lorenz's model, similar to the work by Roebber (1995) or Gottwald (2021), where the variation of AMOC is driven by the chaotic dynamical system. The thermal mode and

the reverse haline mode correspond to different equilibrium states of the AMOC. For simplicity, we will refer to these different states as the stronger AMOC (on-state) and the weaker AMOC (off-state) in simple conceptual models (e.g., Weijer et al., 2019).

Data assimilation that combines a model with observed data is a feasible approach to study the multi-equilibria of AMOC
given the situation described above. A popular data assimilation scheme is the Kalman filter (Kalman, 1960; Kalman and Bucy, 1961). The main idea is to adjust the model predictions according to the observational data to obtain an optimal estimation of model states. Combining the Kalman filter with the idea of ensemble prediction, the ensemble Kalman filter (EnKF) uses ensemble samples of system states to estimate the background error covariance (e.g., Evensen, 1994). As a variant of EnKF, the ensemble adjustment Kalman filter (EAKF) derives a linear operator from the product of the observational distribution and
the prior distribution of the model state to update the model ensemble (Anderson, 2001). EAKF has been applied to climate models to have developed fully coupled data assimilation systems (e.g., Zhang et al., 2007; Liu et al., 2014b). Tardif et al. (2014) implement data assimilation with EnKF to recover the AMOC with observations in a low-order coupled atmosphere-ocean climate model. They mainly explore the value of data assimilation for the initialization of the AMOC, while the effect of parameter errors in AMOC simulations needs further discussion. As another class of ensemble-based assimilation methods,
particle filters, unlike the EnKF, are applicable to non-Gaussian probability distributions (e.g., Gordon et al., 1993; van Leeuwen, 2009). A mixture-based implicit particle method is presented and could detect transitions in an example with multiple attracting states (Weir et al., 2013a). However, the particle filter is plagued by the curse of dimensionality as the system dimension increases (Snyder et al., 2008; Carrassi et al., 2018).

The method of parameter estimation is based on the theory of data assimilation, i.e. information estimation theory, or
120 *filtering theory* (e.g., Jazwinski, 1970). Research on the use of observations to estimate model parameters has attracted extensive attention and has produced encouraging results in the literature (Annan et al., 2005; Aksoy et al., 2006a, 2006b; Hansen and Penland, 2007; Kondrashov et al., 2008; Hu et al., 2010). Based on EAKF, a data assimilation scheme for enhanced parameter correction is designed to improve parameter estimation using observations (Zhang et al., 2012). Zhao et al. (2019) perform this scheme in a simple AMOC box model, and the model parameters are successfully optimized when the model
errors are caused by only erroneously set parameters. Although the AMOC regime transition is not addressed in their study, their exploration of model sensitivities regarding parameters serves as a guideline for our research. Many efforts have been made to advance the application of data assimilation and parameter estimation in nonlinear systems having multiple equilibrium states (e.g., Miller et al., 1994, 1999; Khalil et al., 2009; Weir et al., 2013b; Bisaillon et al., 2015). Although numerical simulations of the AMOC eventually exhibit multiple equilibria, the AMOC is not an explicit model variable; rather,
it is derived from model variables such as atmospheric wind, ocean temperature and salinity. Instead of adjusting AMOC directly, the model states are adjusted through data assimilation. When constraining model parameters by observational information, the parameters that constantly vary with observations may provide more diversity in the physical processes involved with AMOC regime transition, so that the model can simulate more AMOC transition paths.

Here we present a method for improving the modeling of AMOC multi-equilibria. The new method is shown to simulate the AMOC transition between different equilibrium states accurately in two simple coupled models, the first combining a three-box overturning simulation model with a five-variable simple climate model, and the second with clearer physical meaning. Then, we apply EAKF to both AMOC models to establish an ensemble coupled model data assimilation and parameter estimation (CDAPE) system, respectively. Within a "twin" experiment framework, the "observation" information, which is from the assimilation model simulation, is used to adjust the parameters of the model, thereby constraining the paths of transition between different AMOC equilibrium states, so that the path simulated by the model is close to the "real" path.

This paper is organized as follows. After the introduction, the methodology is described in Sect. 2, including a general proposition of optimizing multi-equilibrium transition path of AMOC, the EAKF algorithm, and the design of twin experiments used throughout this study. Section 3 begins with a description of the three-box and five-variable models, and their combination to simulate regime transitions of the AMOC, and then describes the optimization of the trajectory for the multi-equilibrium transition by CDAPE. In Sect. 4, the method of capturing regime transitions by CDAPE is applied to a similar simple model with a more explicit physical meaning. Finally, the summary and discussions are given in Sect. 5.

## 2 Methodology

### 2.1 Simulation and optimization of AMOC regime transitions

The AMOC, as part of the thermohaline circulation, consists mainly of warmer and saltier water flowing from low to high latitudes in the upper ocean of the Atlantic, colder North Atlantic Deep Water (NADW) flowing southward in the deep ocean, and the corresponding upwelling and downwelling currents (Fig. 1a). Multiple equilibria exist in the system, for example, including the thermal mode (active AMOC or on-state) and the reverse haline mode (weak AMOC or off-state). The regime transitions of AMOC are simulated in simple idealized box models and complex ocean general circulation models. There are many influencing factors involved in the model, such as wind-driven gyre and freshwater flux, etc, and their variations will result in different states for AMOC. Their specific values and parameterization schemes are often designed with respect to the model states $x$.

Generally, an ocean-atmosphere coupled model which contains complex physical processes, can be generally expressed as $\frac{\partial x}{\partial t} = F(x, \beta)$, where the model states (vector $x$) include the atmospheric and oceanic states. The model contains a set of fixed standard parameters $\beta$, and the values of $\beta$ might be subject to errors, limited by incomplete understanding of physical processes and inadequate modeling experience and measurements, etc. The state of AMOC can be derived from the model state as $MOC = f(x)$. Fixed-value parameters in a single model may result in simulations that do not cover multiple equilibria in the real system. On the other hand, errors in the model parameters can result in inconsistent AMOC regime transition between the model simulation and reality. Focusing on these issues, our study explores whether it is possible to project observational

information onto model states and parameters so that model simulation behavior fits to realistic multiple equilibrium states and capture regime transitions by data assimilation and parameter estimation.

As shown in Fig. 1b, the blue line is the time series of *MOC*, representing the AMOC from the model control simulation, computed from the model states $x$ by the relation $f$, and the dashed red line represents the "real" multi-equilibrium transition path. The dashed black line is a division line between two equilibrium states. The multi-equilibrium transition path (blue line) from the simulation control model with fixed parameters $\beta$ is restricted to one equilibrium state, while the "real" transition path is more flexible in transforming between two states. For shorter timescales (at most multi-decadal timescales), limited by the scarcity of direct observations of AMOC, information on the AMOC variations can only be obtained indirectly, through direct observations $y$ in real Earth system, such as atmospheric wind, ocean temperature and salinity, etc. For longer timescales (centurial and millennial timescales), the observations of AMOC can be derived from the paleoclimate records $y$. The red "+" signs in Fig. 1b, derived from the direct observations $y$, represent the indirect "observations" of the AMOC sampled from reality. The observations $y$ are projected onto the model parameters $\beta$ by CDAPE (the red arrow in Fig. 1b) so that $\beta$ evolves over time with observation-dependent trend. Since the varying parameters allow the physical process of the model to be more flexible and the parameters $\beta$ constrained by observation $y$ gradually approach their true values, the model CDAPE simulation (purple line) results in a more realistic AMOC multi-equilibria (the blue arrow). To explore how likely this idea is to be realized, we attempt to capture AMOC regime transitions, in a conceptual model reflecting the characteristic of multi-equilibria and a more complex model with simple physical processes, and even planned in a more complex ocean-atmosphere coupled model representing the real Earth system.

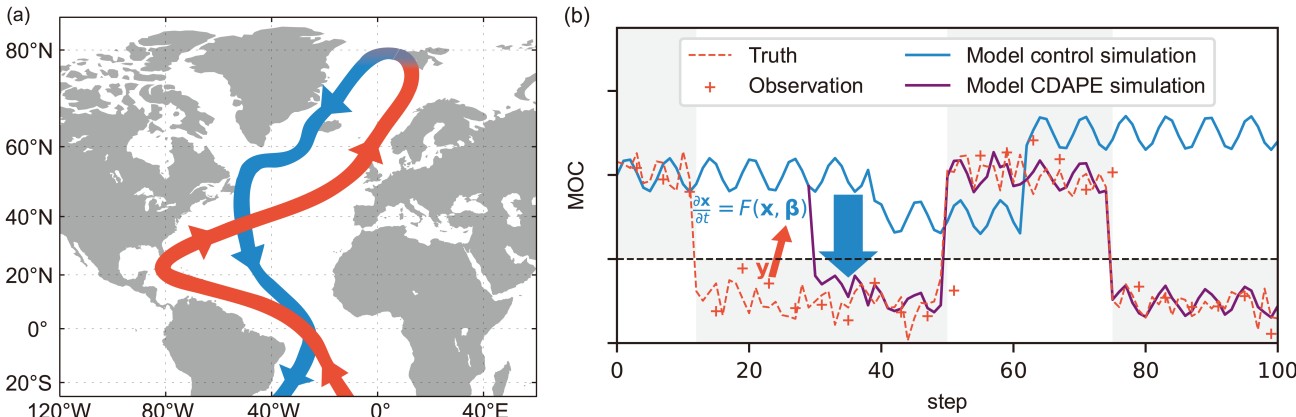

**Figure 1.** A schematic illustration of capturing AMOC regime transitions by projecting observational information to model states and parameters, by a) the illustration of AMOC consisting mainly of upper warmer and saltier northward flow (red) and deep cold southward flow (blue), and b) the time series of the "observations" ("+" signs) of the Ocean, the values of model control simulation (blue line) and observation-constrained simulation by coupled data assimilation and parameter estimation (CDAPE). The red arrow denotes that the "observations" sampled from the truth (dashed red line) are projected onto the model states $x$ and parameters $\beta$ at nearly step 30. The purple line represents the model simulation results after CDAPE. The blue arrow represents the process of capturing regime transitions. The backgrounds with color of light gray correspond to the "observed" multi-equilibrium transition path. The dashed black line is a division line between different equilibrium states of the AMOC.

**2.2 Coupled model data assimilation and parameter estimation (CDAPE)**

The ensemble adjustment Kalman filter (Anderson, 2001) is used for data assimilation and parameter estimation in this study. The basic process of the two-step EAKF (Anderson, 2003; Zhang and Anderson, 2003; Zhang et al., 2007) is to project the observational increment onto model states (relevant parameters) by calculating the error covariance between the prior ensemble of the model variable (parameter) and the model-estimated ensemble. The core of the two-step EAKF is to calculate the increment of each state variable by a local least squares fit (linear regression), and the calculation of the observational increment is related to the scalar application of the equations of EAKF (Anderson, 2003). All observations at time $t$ have the observation value $\boldsymbol{y^o}$ (in $N_{obs}$ dimensions). For a single observation $y_k^o$ at the $k$-th observation location ($k = 1 \sim N_{obs}$), the standard deviation of observational error is $\sigma^o$ (assumed to be Gaussian). The model states are mapped onto the observational space by applying a linear interpolation, and then the prior (model-estimated) ensemble of the $k$-th observation $\boldsymbol{y_k^p}$ (in $N_{ens}$ dimensions) can be obtained. $y_{k,i}^p$ is the $i$-th prior ensemble member of the $k$-th observation. The ensemble mean and standard deviation are $\bar{y}_k^p$ and $\sigma_k^p$, respectively.

The first step is to compute the observational increment of the $k$-th observation ($k = 1 \sim N_{obs}$). The observational increment $\Delta y_{k,i}^o$ for the $i$-th ensemble member ($i = 1 \sim N_{ens}$) is formulated by

$$\Delta y_{k,i}^o = \bar{y}_k^u + \Delta y_{k,i}' - y_{k,i}^p , \qquad (1)$$

where $\bar{y}_k^u$ is the posterior ensemble mean of the $k$-th observation, representing the shift of the ensemble mean induced by this observation, $\Delta y_{k,i}'$ is the updated ensemble spread of the $k$-th observation, representing the reshaping of the model ensemble. They are respectively computed by

$$\bar{y}_k^u = \frac{(\sigma^o)^2}{(\sigma^o)^2 + (\sigma_k^p)^2} \bar{y}_k^p + \frac{(\sigma_k^p)^2}{(\sigma^o)^2 + (\sigma_k^p)^2} y_k^o , \text{ and}$$

$$\Delta y_{k,i}' = \sqrt{\frac{(\sigma^o)^2}{(\sigma^o)^2 + (\sigma_k^p)^2}} \left(y_{k,i}^p - \bar{y}_k^p\right) , \qquad (2)$$

where the first equation shows whether the ensemble mean shifts closer to the prior model ensemble mean $\bar{y}_k^p$ or the observation value $y_k^o$, and whether it is $\bar{y}_k^p$ or $y_k^o$ depends on which has the smaller variance. The second equation denotes that the prior probability density function is squashed by a new observation.

The second step is to distribute the observational increments $\Delta y_{k,i}^o$ on to the related model states $\boldsymbol{x}$ (a matrix of size $N_{ens} \times N_{state}$) and this assimilation process can be expressed as

$$\Delta x_{j,i} = \frac{cov(x_j, \boldsymbol{y_k^p})}{(\sigma_k^p)^2} \Delta y_{k,i}^o , \qquad (3)$$

where $\Delta x_{j,i}$ is the contribution of the $k$-th observation to the $i$-th ensemble member of the $j$-th model variable $x_{j,i}$ ($j = 1 \sim N_{state}$). $cov(\boldsymbol{x_j}, \boldsymbol{y_k^p})$ is the error covariance between the prior ensemble of the $j$-th model variable $\boldsymbol{x_j}$ (in $N_{ens}$ dimensions)

and the prior (model-estimated) ensemble of the $k$-th observation $\boldsymbol{y_k^p}$ (in $N_{ens}$ dimensions), and is calculated as $cov(\boldsymbol{x_j}, \boldsymbol{y_k^p}) =$

$\frac{\sum_{i=1}^{N_{ens}}(x_{j,i}-\bar{x}_j)\left(y_{k,i}^p-\bar{y}_k^p\right)}{N_{ens}}$, where $\bar{x}_j$ is the ensemble mean of $j$-th model variable.

The model parameters are fixed when parameter estimation is not performed. The parameters vary with observational information by parameter estimation. The core of the parameter estimation is to obtain the increment of the estimated parameter by a linear regression that is based on the error covariance between the prior parameter ensemble and the state ensemble (Anderson, 2001, 2003). The error covariance used in regression is flow dependent and temporally varying (Zhang and

225 Anderson, 2003). Therefore, for the model parameter estimation, the observational increments are distributed onto a relevant parameter and the equation is

$$\Delta\beta_{j,i} = \frac{cov(\boldsymbol{\beta_j}, \boldsymbol{y_k^p})}{(\sigma_k^p)^2}\Delta y_{k,i}^o \,, \tag{4}$$

where $\Delta\beta_{j,i}$ is the contribution of the $k$-th observation to the $i$-th ensemble member of the $j$-th parameter being estimated, called $\beta_{j,i}$ ($j = 1 \sim N_{para}$). $cov(\boldsymbol{\beta_j}, \boldsymbol{y_k^p})$ is the error covariance between the prior ensemble of the $j$-th model parameter $\boldsymbol{\beta_j}$ (in $N_{ens}$

dimensions) and the prior (model-estimated) ensemble of the $k$-th observation $\boldsymbol{y_k^p}$ (in $N_{ens}$ dimensions), and is calculated as $cov(\boldsymbol{\beta_j}, \boldsymbol{y_k^p}) = \frac{\sum_{i=1}^{N_{ens}}(\beta_{j,i}-\bar{\beta}_j)\left(y_{k,i}^p-\bar{y}_k^p\right)}{N_{ens}}$, where $\bar{\beta}_j$ is the ensemble mean of $j$-th model parameter being optimized.

Since the model parameters do not have dynamically supported internal variability, the ensemble spread of an estimated parameter will decrease rapidly after several time steps of parameter estimation. In other words, the model parameters are not dynamical variables, which leads to a progressively decreasing ensemble variance of a parameter being estimated. The

235 parameter ensemble will not be adjusted by new observations if the ensemble spread is too small, so the inflation scheme of the prior parameter ensemble is necessary for the parameter estimation. A typical inflation scheme is "conditional covariance inflation" method (Aksoy et al., 2006a). A predefined standard deviation is first chosen empirically as critical value in this scheme. Then the parameter spread is adjusted back to it when the standard deviation of the parameter ensemble is smaller than this critical value. To further improve the signal-to-noise ratio of parameter estimation, Zhang (2011a) introduced an

240 inflation scheme based on model sensitivity with respect to the parameter being estimated. In this inflation scheme, the inflation amplitude of a parameter ensemble is inversely proportional to the sensitivity. It is formulated as $\tilde{\beta}_{j,i} = \bar{\beta}_j + max\left(1, \frac{\alpha_0\sigma_0}{\sigma_j\sigma_t}\right)(\beta_{j,i} - \bar{\beta}_j)$, where $\tilde{\beta}_{j,i}$ denotes the inflated version of the $i$-th ensemble member of the $j$-th parameter being estimated, $\sigma_0$ and $\sigma_t$ are the prior ensemble spreads of this parameter at the initial time and time $t$, $\alpha_0$ is a constant tuned by a trial-and-error procedure (e.g., Wu et al., 2016), and $\sigma_j$ is the sensitivity of the model state with regard to $j$-th parameter. This

indicates that if the prior ensemble spread of $j$-th parameter is smaller than $\frac{\alpha_0}{\sigma_j}$ times the initial spread, it will be enlarged to this amount (e.g., Wu et al., 2012; Han et al., 2014; Zhao et al., 2019). In this study, considering that the inflated parameter ensemble will influence state variables, for the simplicity and convenience of computation and comparison, no inflation is

applied to the model state ensemble, as in Han et al. (2014), Yu et al. (2017) and Zhao et al. (2019). The inflation scheme is only used for parameter estimation.

## 2.3 Experimental design

To show the contribution of data assimilation and parameter estimation to capture AMOC regime transitions, a twin experiment containing a truth model and assimilation models is designed. Since this study focuses on the effect of parameters on multiple equilibria, it is assumed that the model bias is originated only from the incorrectly set parameter. The parameter in the truth model is set to the truth value $\beta_0$, and the simulation results represent the real state of AMOC in reality. Similar to the observation process in reality, the observations $y$ are obtained by superimposing white noise on the real state and sampling at a certain frequency. The assimilation models differ from the truth model only in the parameter values, with the same initial conditions and other aspects such as the differential scheme. The parameter in the $i$-th assimilation model is assumed to be incorrectly guessed as $\beta_i$, and all $\beta_i$ in the assimilation models have the mean of $\beta_m$ ($\beta_m \neq \beta_0$) and the variance of $\beta_v$. The role of data assimilation is shown by the model states constrained by the observations $y$, and furthermore, the role of parameter estimation is shown by the model states obtained after the parameters are constrained by the observations $y$.

## 3 Capturing regime transitions by observation-constrained model parameters in a conceptual MOC model

### 3.1 The MOC3B-5V model

#### 3.1.1 A three-box MOC model

In the classic two-box model of Stommel (1961), a buoyancy constraint on the thermohaline circulation was present. Following the energy-constraint approach, the thermohaline circulation is driven and maintained by mechanical energy so that buoyancy constraint is replaced by an energy constraint (Guan and Huang, 2008). On this basis, considering the effect of wind-driven gyre, a three-box model is formulated (Shen et al., 2011).

The three-box model used in this study has an upper box representing the mid and low latitude surface ocean, a pole box representing the high-latitude ocean, and a lower box representing the mid and low latitude deep ocean, as illustrated in Fig. 2. The three-box model is designed with two different modes which are the thermal mode (driven by temperature) and the haline mode (driven by salinity). In the thermal mode, water flows from the pole box, passing the lower box, then flowing into the upper box by upwelling, and finally returning to the pole box (solid arrows in the boxes), while the circulation is reversed in the haline mode (dotted arrows). The horizontal and vertical water flow are represented by the terms $u$ and $v$, respectively (more details are given in Shen et al., 2011).

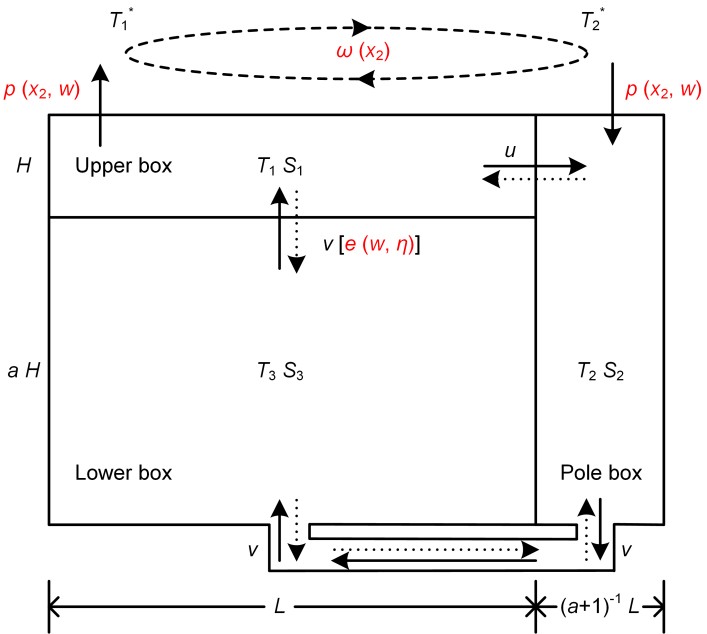

**Figure 2.** Schematic illustration of a three-box model for the thermohaline circulation. The volume of upper box is the same as the volume of pole box, and their volumes are both a multiplication of $a^{-1}$ and the volume of lower box. $L$ and $H$ are the width and depth of upper box, $T_i$ and $S_i$ represent the temperature and salinity in box $i$. The upper boundary conditions are a temperature relaxation toward the specified reference temperature $T_1^* = T_0^*$, $T_2^* = 0$ and a freshwater flux $p$. $\omega$ represents the wind-driven gyre between the upper box and pole box. $u$ is

280 the horizontal water flow and $v$ is the overturning rate. The red terms represent the main influencing factors of $v$ and multiple equilibria, and which variables in the five-variable model they will be combined with. The solid (dotted) arrows in the three boxes indicate the thermal (haline) mode with circulation flowing clockwise (counterclockwise).

The heat balance equations and the salinity balance equations in each box are established firstly. By introducing the nondimensional variables, after derivation, the simple non-dimensional ordinary differential equations are finally obtained as

follows:

$$\dot{T}_1 = v_t T_3 - u_t T_1 + T_0^* - T_1 + \omega(T_2 - T_1)$$

$$\dot{T}_2 = u_t T_1 - v_t T_2 - \frac{1}{a+1}T_2 + \omega(T_1 - T_2)$$

$$a\dot{T}_3 = v_t(T_2 - T_3)$$

$$\dot{S}_1 = v_t S_3 - u_t S_1 + \omega(S_2 - S_1)$$

$$\dot{S}_2 = u_t S_1 - v_t S_2 + \omega(S_1 - S_2)$$

$$a\dot{S}_3 = v_t[(a+2)S_0^* - (a+1)S_3 - S_1],\qquad(5)$$

where $T_i$ and $S_i$ represent the temperature and salinity in box $i$, an overdot denotes time tendency, $T_0^*$ and $S_0^*$ are the mean temperature and mean salinity of the box model ocean, the subscript $t$ stands for the thermal mode, $\omega$ represents the wind-driven gyre, and $p$ represents the freshwater flux. The non-dimensional continuity equation is

295 $$u_t = v_t - p.\qquad(6)$$

Under the energy constraint, the scale of overturning rate in the three-box model satisfies

$$v_t = \frac{e}{\rho_0 \alpha (T_1 - T_2) - \rho_0 \beta (S_1 - S_2)} \,, \tag{7}$$

where $e$ represents the strength of the external source of mechanical energy sustaining mixing, $\rho_0$ is the mean density of the model ocean, $\alpha$ is the thermal expansion coefficient, and $\beta$ is the saline expansion coefficient.

300  In the haline mode, the influence of wind-driven gyre is the same as it is in the thermal mode, but the circulation in three boxes is reversed. The governing equations for the haline mode also follow the study of Shen et al. (2011). They didn't show those equations, but only described them briefly. In this paper, to describe the construction of MOC3B-5V more clearly later, those equations are shown here. Accordingly, the non-dimensional equations in each box are

$$\dot{T}_1 = u_s T_2 - v_s T_1 + T_0^* - T_1 + \omega(T_2 - T_1)$$

305 $$\dot{T}_2 = v_s T_3 - u_s T_2 - \frac{1}{a+1} T_2 + \omega(T_1 - T_2)$$

$$a\dot{T}_3 = v_s(T_1 - T_3)$$

$$\dot{S}_1 = u_s S_2 - v_s S_1 + \omega(S_2 - S_1)$$

$$\dot{S}_2 = v_s S_3 - u_s S_2 + \omega(S_1 - S_2)$$

$$a\dot{S}_3 = v_s[(a+2)S_0^* - (a+1)S_3 - S_2] \,, \tag{8}$$

310 where the subscript $s$ stands for the haline mode. The non-dimensional continuity equation is

$$u_s = v_s + p \,. \tag{9}$$

The overturning rate's $v_s$ is formulated by

$$v_s = \frac{e}{\rho_0 \beta (S_1 - S_2) - \rho_0 \alpha (T_1 - T_2)} \,. \tag{10}$$

  Equations (5)-(7) are governing equations for the thermal mode, and Equations (8)-(10) are governing equations for the

315 haline mode of the thermohaline circulation in a hemisphere three-box model. The time derivatives in Eqs. (5) and (8) are set to be zero, and then the governing equations for the thermal mode or the haline mode are solved, respectively. Equations (5)-(7) have one stable solution, and Equations (8)-(10) have one stable solution and one unstable solution. Hence, the three-box model has a total of three mathematical solutions. This result obtained by solving the equations could be found in Shen et al. (2011). Similar equations for the thermal and haline modes could be found in Guan and Huang (2008) for Eq. (1) (thermal mode) and Eq. (2) (haline mode), and in Shen and Guan (2015) for Eqs. (1)-(6) (thermal mode) and Eqs. (7)-(9) (haline mode).

320 The overturning rate ($v$) and multiple equilibria are affected by energy constraint $e$, freshwater flux $p$, and wind-driven gyre $\omega$. The haline mode will switch to the thermal mode when $e$ or $\omega$ is increased or $p$ is decreased beyond the critical value (Shen and Guan, 2015).

### 3.1.2 A 5-variable conceptual climate model

325 Lorenz (1963) proposed a simple model with only three variables to simulate the chaotic characteristics of the atmosphere, where $x_1$ is proportional to the intensity of the convective motion, $x_2$ is proportional to the temperature difference between the ascending and descending currents, and $x_3$ is proportional to the distortion of the vertical temperature profile from linearity.

However, its three variables only reflect the process of atmospheric convection, and they cannot represent the interaction of the atmosphere and ocean, as well as the low-frequency nature of climate evolution. On this basis, two ocean variables that represent the slab ocean variable and the deep ocean pycnocline anomaly are added and coupled with the chaotic "atmosphere" to simulate the interactions between the atmosphere and the ocean (Zhang et al., 2012) as well as the upper and deep oceans (Zhang, 2011a, 2011b). The model equations are:

$$\dot{x}_1 = -\sigma x_1 + \sigma x_2$$

$$\dot{x}_2 = -x_1 x_3 + (1 + c_1 w)\kappa x_1 - x_2$$

$$\dot{x}_3 = x_1 x_2 - b x_3$$

$$O_m \dot{w} = c_2 x_2 + c_3 \eta + c_4 w \eta - O_d w + S_m + S_s \cos(2\pi t / S_{pd})$$

$$\Gamma \dot{\eta} = c_5 w + c_6 w \eta - O_d \eta \,, \tag{11}$$

where $x_1$, $x_2$, and $x_3$ are the high-frequency variables that represent the atmosphere, $w$ and $\eta$ are the low-frequency variables that conceptually simulate the simple variation characteristics of the upper ocean and the deep ocean, respectively.

The original $\sigma$, $\kappa$, and $b$ sustain the chaotic nature of the atmosphere. The coupling between the fast atmosphere and the slow ocean is reflected by $c_1$ and $c_2$. The coefficient $c_1$ represents the oceanic forcing on the atmosphere and $c_2$ represents the atmospheric forcing on the ocean. To ensure that the time scale of the ocean is slower than the atmosphere, the heat capacity $O_m$ must be much larger than the damping rate $O_d$. For $w$, the parameters $S_m$ and $S_s$ define the magnitudes of the annual mean and seasonal cycle of the imposed external forcing, and the period of the seasonal cycle is defined by $S_{pd}$. The interactions and nonlinear interactions of upper ocean and deep ocean are represented by coefficients $c_3$, $c_4$, $c_5$, and $c_6$. The terms $c_3\eta$ and $c_4 w \eta$ ($c_5 w$ and $c_6 w \eta$) represent the linear exchange flux and the nonlinear role from deep (upper) ocean to upper (deep) ocean. The ratio of the constant of proportionality $\Gamma$ and $O_d$ determines the time scale of variation of $\eta$. The standard values of these parameters in the model are set to $(\sigma, \kappa, b, c_1, c_2, O_m, O_d, S_m, S_s, S_{pd}, \Gamma, c_3, c_4, c_5, c_6) = (9.95, 28, 8/3, 10^{-1}, 1, 1, 10, 10, 1, 10, 100, 10^{-2}, 10^{-2}, 1, 10^{-3})$.

### 3.1.3 The 3-box MOC model coupled with the 5-variable model (MOC3B-5V)

The construction of the MOC3B-5V model starts with the three-box model of the previous study of Shen and Guan (2015), including the non-dimensional temperature and salinity differential equations, the continuity equations, and the equation for the overturning rate (Eqs. 5-10). To obtain the time series of overturning rate and simulate the AMOC transition between different equilibrium states in the time series, we no longer set the time derivatives in Eqs. (5) and (8) to be zero, instead using a leapfrog time differencing scheme to forward the temperature and salinity so as the overturning rate. Although an unstable equilibrium can be obtained by solving the equations, a small perturbation on the solution will grow exponentially (Shen et al., 2011), so it is rather than a physical solution. In this study, the equilibrium states obtained by using the time differencing scheme are all stable solutions.

To test the feasibility of the time differencing scheme, the values of $e$, $p$, $\omega$ in the three-box model are changed respectively from small to large, and the overturning rate is calculated when the temperature and salinity in the three boxes are almost steady, which means that the AMOC reaches a quasi-equilibrium state. By using a leapfrog time differencing, the three-box model is first spun up for $10^5$ TUs (Time units, 1 TUs = 100 steps) starting from $(T_1, T_2, T_3, S_1, S_2, S_3) = (20.0, 0.0, 15.0, 35.5, 35.0, 34.5)$ with the values of relevant parameters described in the previous study (Shen and Guan, 2015). The initial values of temperature and salinity at the equilibrium state are obtained. Then the value of $e$ (from 0.0 to $3.0 \times 10^{-7}$ kg m$^{-2}$ s$^{-1}$), $p$ (from 1.0 to 0.0 m yr$^{-1}$) or $\omega$ (from 0.0 to 5.0 Sv, 1 Sv = $10^6$ m$^3$ s$^{-1}$) is changed within a certain range. Each time it changes, the three-box model is integrated for another 500 TUs for spin-up to reach an equilibrium state, and the overturning rate corresponding to different values of $e$, $p$, $\omega$ is calculated. To distinguish the overturning rate in the haline mode from that in the thermal mode, the overturning rate in the haline mode can be represented by $-v_s$, which means that the circulation is reversed.

The results are consistent with previous results from the research on model stability in Shen et al. (2011). Figure 3a shows the effects of $e$ on the circulation in the three-box model. The corresponding threshold of $e$ exists in the haline mode. When $e$ is less than the critical value, the overturning rate is less than zero. When the value of $e$ is increased beyond the threshold, the haline mode switches to the thermal mode. Similarly, when $p$ is decreased beyond the corresponding critical value (in Fig. 3b), or when $\omega$ is increased beyond the corresponding critical value (in Fig. 3c), the AMOC transition from the haline mode to the thermal mode.

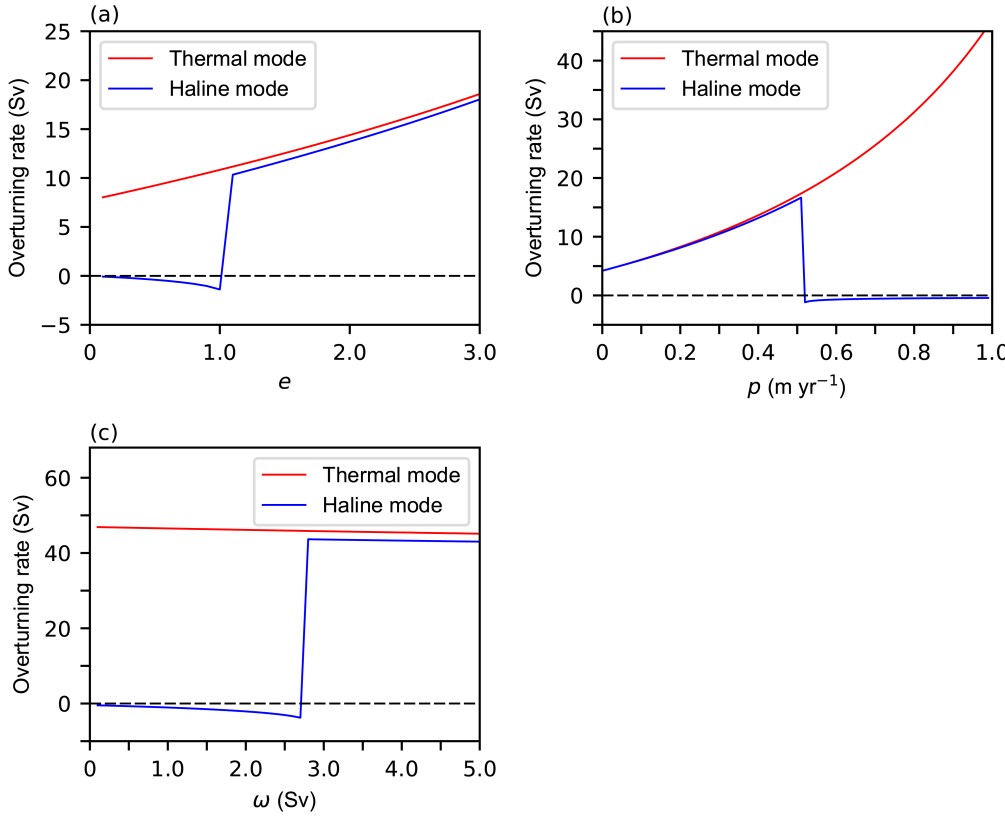

**Figure 3.** Variations of overturning rates in the space of a) energy constraint parameter $e$, b) freshwater flux $p$ and c) wind-driven circulation $\omega$ in the three-box model with a) $p = 0.5$ m yr$^{-1}$ and $\omega = 1.5$ Sv, b) $e = 2.5 \times 10^{-7}$ kg m$^{-2}$ s$^{-1}$ and $\omega = 0.0$ Sv, c) $e = 2.5 \times 10^{-7}$ kg m$^{-2}$ s$^{-1}$ and $p = 1.0$ m yr$^{-1}$. The red line indicates the stable thermal mode and the blue line indicates the stable haline mode. The dashed black line is a division line between the thermal mode equilibrium state and the haline mode equilibrium state.

To simulate the transition between different states of AMOC and achieve the shift from the thermal mode to the haline mode, the non-dimensional differential equations for temperature and salinity balance are adjusted by adding a term $Q_a$ which represents an additional freshwater flux from the atmosphere to the upper box and the pole box. Similar to the parameterization scheme in Roebber (1995), a simple and more idealized parameterization scheme for $Q_a$ is devised, which assumes that the additional freshwater flux from the atmosphere to the ocean is divided into mean transport components and transient components. The transient components are assumed to be linearly related to $x_2$. Then, the term $Q_a$ can be defined as $Q_a = Q_0 + \alpha_0 x_2$, where $Q_0$ and $\alpha_0$ are constants. Since the additional freshwater flux $Q_a$ should be much smaller than the freshwater flux $p$, the values of $Q_0$ and $\alpha_0$ are set to 0.02 and 0.000125 considering the magnitude and variation of $p$ and $x_2$. The calculation result of the overturning rate in the thermal mode (denoted by $v_t$) is different from that in the haline mode (denoted by $v_s$). To unify Eqs. (5)-(7) for the thermal mode and Eqs. (8)-(10) for the haline mode, "$-v_s$" and "$-u_s$" are introduced in the haline mode. Thus, $v$ or $u$ greater (less) than zero represents the thermal (haline) mode with circulation flowing clockwise (counterclockwise) in the three boxes in Fig. 2.

Hence, the non-dimensional differential equations are

$$\dot{T}_1 = v[\theta(v)T_3 + \theta(-v)T_1] - u[\theta(v)T_1 + \theta(-v)T_2] + T_0^* - T_1 + \omega(T_2 - T_1)$$

$$\dot{T}_2 = u[\theta(v)T_1 + \theta(-v)T_2] - v[\theta(v)T_2 + \theta(-v)T_3] - \frac{1}{a+1}T_2 + \omega(T_1 - T_2)$$

$$a\dot{T}_3 = v[\theta(v)T_2 + \theta(-v)T_3] - v[\theta(v)T_3 + \theta(-v)T_1]$$

$$\dot{S}_1 = v[\theta(v)S_3 + \theta(-v)S_1] - u[\theta(v)S_1 + \theta(-v)S_2] + \omega(S_2 - S_1) + Q_a$$

$$\dot{S}_2 = u[\theta(v)S_1 + \theta(-v)S_2] - v[\theta(v)S_2 + \theta(-v)S_3] + \omega(S_1 - S_2) + Q_a$$

$$a\dot{S}_3 = v[\theta(v) - \theta(-v)][(a+2)S_0^* - (a+1)S_3] - v\theta(v)S_1 + v\theta(-v)S_2 , \qquad (12)$$

where the function $\theta(x)$ is a step function, which has the value 1 for a positive argument and the value zero otherwise. Through this function, we can represent the different circulation given by different sign of $v$. The continuity equation is

$$u = v - p , \qquad (13)$$

and the overturning rate can take a form as

$$v = \frac{e}{\rho_0 \alpha(T_1 - T_2) - \rho_0 \beta(S_1 - S_2)} , \qquad (14)$$

so that the sign of $v$ will represent where equilibrium state the AMOC is. The intensity of overturning rate and the state of AMOC are mainly affected by $e, p$, and $\omega$ in the circulation control equations. A similar AMOC box model with many switches could be found in Castellana et al., (2019).

In reality, the circulation intensity and the AMOC state are affected by many factors, such as mechanical energy, which is directly used to sustain the vertical mixing in stratification, freshwater flux, and wind-driven circulation. These factors change irregularly in the earth system. To make the variations of $e, p$, and $\omega$ in the model have chaotic characteristics, which is similar to reality, these three influencing factors and the five-variable model are combined. Since the energy constraint $e$ is related to the upper ocean and the deep ocean, the freshwater flux $p$ is related to the atmosphere and the upper ocean, and the wind-driven circulation $\omega$ is directly related to the atmosphere, it is possible to conceptually idealize a simple equation for the relationship between $e, p, \omega$ with $x_2, w, \eta$.

The energy constraint $e$ reflects the strength of the external mechanical energy that sustains mixing, the main sources of which are the energy provided by the wind and tidal dissipation. In this process, kinetic energy is converted to potential energy through turbulence and internal waves (Huang, 2004). Such external mechanical energy is estimated to be about 2 TW (terawatts), with about 1.2 TW as the contribution of wind to mixing, including the generation of internal waves in the surface ocean (Munk and Wunsch, 1998), and the energy from the wind can radiate throughout the ocean (Wunsch and Ferrari, 2004). Besides, a previous study has estimated the energy provided by wind at 1 TW, which is also about half of the total external energy to sustain mixing (Wunsch, 1998). The other half of the total energy comes mainly from the tidal dissipation in the deep ocean, and to a lesser extent from the interactions of the eddy with the ocean bottom topography (Wunsch and Ferrari, 2004; Kuhlbrodt et al., 2007). The energy parameter $e$ varies continuously with climate conditions, but it is difficult to establish the connection between them accurately, due to the large uncertainty in the estimation of these energy sources (Guan and Huang, 2008).

For the idealized three-box model coupled with the simple conceptual climate model, the energy constraint $e$ can only be conceptually constructed by approximate estimation. However, this paper focuses primarily on capturing regime transitions of the AMOC, so it will not be affected by the inaccurate energy constraint $e$ that is conceptually established. The main sources of external energy to sustain mixing are tide and wind, so $e$ is defined as $e = E_t + E_w$, where $E_t$ represents the kinetic energy originating from the abyssal tidal flow and $E_w$ represents the energy from wind contribution to the ocean. $E_t$ is primarily

associated with the deep ocean, so the equation is simply established as $E_t = a_1 (\eta + b_1)$. Since the wind affects the upper ocean directly and radiates throughout the ocean through the interaction of the upper ocean with the deep ocean, the equation is established as $E_w = a_2 (w + b_2) + a_3 (w + b_2) (\eta + b_1)$, where $a_1$, $a_2$, $a_3$, $b_1$, $b_2$ are constants to be determined. The range of $e$ has been estimated to be roughly $1 \times 10^{-7}$ to $3 \times 10^{-7}$ kg m$^{-2}$ s$^{-1}$ (Guan and Huang, 2008), and considering that the threshold for the equilibrium state transition is near $1.0 \times 10^{-7}$ kg m$^{-2}$ s$^{-1}$ (in Fig. 3a), the mean value of $e$ is taken to be about $1.5 \times 10^{-7}$ kg m$^{-2}$

s$^{-1}$, with wind and tidal contributing half of the total, respectively. Scaling $w$ and $\eta$ based on the mean and range of variation of them, the values for $a_1$, $a_2$, $a_3$, $b_1$, $b_2$ can be readily derived.

    The freshwater flux mainly consists of river runoff (denoted by $P_r$), evaporation, and precipitation (jointly denoted by $P_{ep}$), so $p$ is formulated by $p = P_r + P_{ep}$. Since $P_r$ is primarily associated with the upper ocean, establish the equation $P_r = a_4 (w + b_2)$. $P_{ep}$ are related to the interaction of the upper ocean with the atmosphere, so the equation $P_{ep} = a_5 (x_2 + b_3) + a_6 (x_2 + b_3) (w$

$+ b_2)$ is established. The river runoff accounts for a major portion of the total freshwater flux in the northern part of the Atlantic (Broecker et al., 1990), and concerning Fig. 3b, where the threshold for the equilibrium state transition is approximately 0.5 m yr$^{-1}$, the values of $a_4$, $a_5$, $a_6$ and $b_3$ can be readily derived. The wind-driven circulation is mainly related to the atmospheric forcing, and the equation is simply established as $\omega = a_7 (x_2 + b_3)$. Based on the fact that the equilibrium state transition point is near 2.5 Sv (in Fig. 3c) and scaling is performed on $x_2$, the values of $a_7$ can be estimated. Then, the relationships between $e$,

$p$, $\omega$ from the three-box model and $x_2$, $w$, $\eta$ (red terms in Fig. 2) are established as follows

$$e = a_1(\eta + b_1) + a_2(w + b_2) + a_3(w + b_2)(\eta + b_1)$$
$$p = a_4(w + b_2) + a_5(x_2 + b_3) + a_6(x_2 + b_3)(w + b_2)$$
$$\omega = a_7(x_2 + b_3) . \qquad (15)$$

    A set of parameter values $(a_1, a_2, a_3, a_4, a_5, a_6, a_7, b_1, b_2, b_3) = (3^{-1}, 10^{-1}, 30^{-1}, 9^{-1}, 800^{-1}, 7200^{-1}, 16^{-1}, -11, -7, 40)$ are used as

simulating the variation of $e$, $p$ and $\omega$ in the air-sea system. Therefore, the three-box model and the five-variable model are combined. The time series of overturning rate can be calculated, which can simulate the transition between different states of AMOC.

### 3.2 Experimental design

For the MOC3B-5V model, the model states (vector $\boldsymbol{x}$) are the five variables, the ocean temperature variables, and the ocean

salinity variables. The physical processes $\frac{\partial \boldsymbol{x}}{\partial t} = F(\boldsymbol{x}, \boldsymbol{\beta})$ are represented by Eqs. (11) and (12), where the influencing factors $e$, $p$, $\omega$ are calculated by Eq. (15). The AMOC states are obtained by Eq. (14). Assuming that there is an error between the true

value of a parameter and the value in the model, a twin experiment framework is set. The MOC3B-5V model is set with standard parameter values, where the standard value of original $\kappa$ in Eq. (11) is 28. Starting from the conditions ($x_1$, $x_2$, $x_3$, $w$, $\eta$) = (0, 1, 0, 0, 0), the model is integrated for 300 TUs to obtain the initial values of the five variables. The initial values of

temperature and salinity at the equilibrium state in three boxes are obtained as described in Sect. 3.1.3. When using observations of $x_1$, $x_2$, $x_3$, and $w$ for estimation, $\kappa = 28$ is the "true" solution of the parameter. The truth model is run forward for 5000 TUs to establish the "truth" (see dashed line in Fig. 4 with $x_2$ as a case). To simulate the observation errors, white noise is added to the true value, and the standard deviations of these observation errors are set to 2 for $x_1$, $x_2$, $x_3$, and 0.5 for $w$. Then, the true value with white noise is sampled at a certain frequency (5 time steps for $x_1$, $x_2$, $x_3$, and 20 time steps for $w$) as

observations. The "+" signs in Fig. 4 show an example of observations ($x_2$).

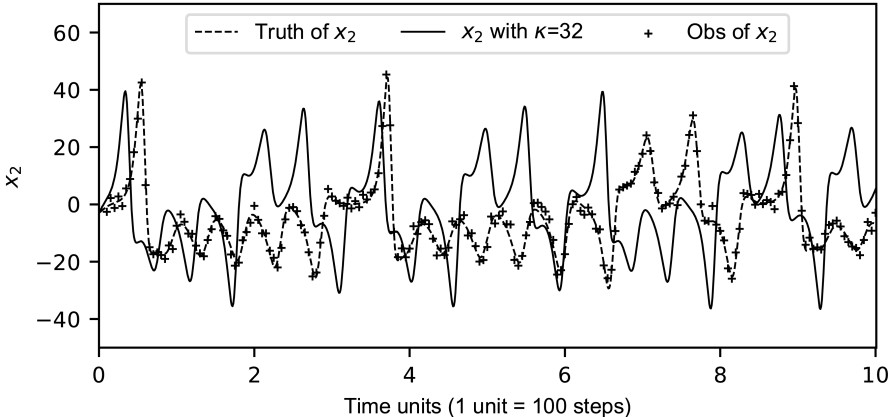

**Figure 4.** Time series of the observations ("+" signs) of $x_2$, and the $x_2$ values of the control model simulation (solid line) with $\kappa = 32$ without data assimilation and parameter estimation. The "observations" are taken from the "truth" simulation (dashed line as the reference) at an interval of 0.05 TU and superimposed by a white noise with a standard deviation of 2.

For the assimilation model, the initial conditions are the same as the above "observation". 20 model ensembles are set with different $\kappa$ to simulate the transitions of AMOC in different models. 20 Gaussian random numbers are drawn for the parameter $\kappa$ to be estimated, with the mean ($\bar{\kappa}_i = 32$) and the guessed standard deviation ($\sigma_0^2 = 0.1$). The 20 models are spun up from the same initial conditions for another 5000 TUs with different values of $\kappa$ and standard values for other model parameters.

**3.3 Sensitivity of model parameters**

Several numerical models have shown that multiple equilibria exist in the thermohaline circulation, but the AMOC regime transitions obtained in different models are different. Changes in freshwater flux, energy constraint, wind-driven circulation, and other factors will cause the AMOC to switch between different equilibrium states. By combining the three-box model with the five-variable model, it is simulated that AMOC switches between different equilibrium states in the time series.

As described in Sect. 3.2, the parameter $\kappa$, which affects the variation of $e$, $p$, and $\omega$ in the model, is erroneously guessed as

32. The errors of AMOC transitions caused by model parameters grow rapidly. For the twenty different ensemble members in

the free model control ensemble simulations, although the values of $\kappa$ are all close to 32 with a small difference and their variance is only 0.1, the simulation results (orange lines in Fig. 5a) are quite different which means that the equilibrium states of AMOC are different. Meanwhile, the path of transition between different equilibrium states is also different and does not converge to the truth (red lines). The overturning rate greater than zero means that AMOC is in the thermal mode. By contrast, 485 the value of overturning rate is negative because of the reversed direction of water flow in the three boxes.

## 3.4 Data assimilation and parameter estimation

AMOC simulation results of twenty ensemble members along different transition paths, which are different from the "observations". To adjust the model to make the simulation results closer to the truth, the "observations" are assimilated into the model. Based on the method in Sect. 2.2, the observational increment and the covariance between the prior ensemble of 490 model variable and the model-estimated ensemble are calculated first, after which each observational increment is applied to Eq. (3) to update the model variable ensemble. Obtain an updated prior ensemble of the variable in preparation for the next cycle of data assimilation. The results show that after data assimilation, although the twenty ensemble members (orange lines in Fig. 5b) are in the same path of transition, where AMOC switches between different equilibrium states at the same pace, their transition paths are still different from the "observational" path (red lines). This is because there is a deviation between 495 the parameter value in the model and its best estimate.

Parameterization can approximate many physics in the model, but the values of parameters are usually estimated roughly by summing up experiences in a large number of experiments. To reduce the error caused by parameter errors between model simulation results and the truth, parameter estimation is performed next. The observational increment is applied to the error covariance between the model-estimated ensemble and the prior parameter ensemble by Eq. (4). Parameter estimation starts 500 at 300th-unit. The result shows that after 300th-unit, the overturning rate in twenty ensemble members all follow the same transition path (orange lines in Fig. 5c), which is the same as the "observational" path (red lines). Meanwhile, the parameter $\kappa$ is adjusted to around the best-estimated value 28 (Fig. 6). As an example of capturing regime transitions of the AMOC, Figure 7 shows the results in one of the twenty ensemble members in the free model control ensemble simulations with or without CDAPE. From Fig. 7, we learned that although the model parameter $\kappa$ is erroneously guessed, constraining $\kappa$ with observational 505 data can change the path of AMOC transition between different equilibrium states. The model deviations are mitigated significantly.

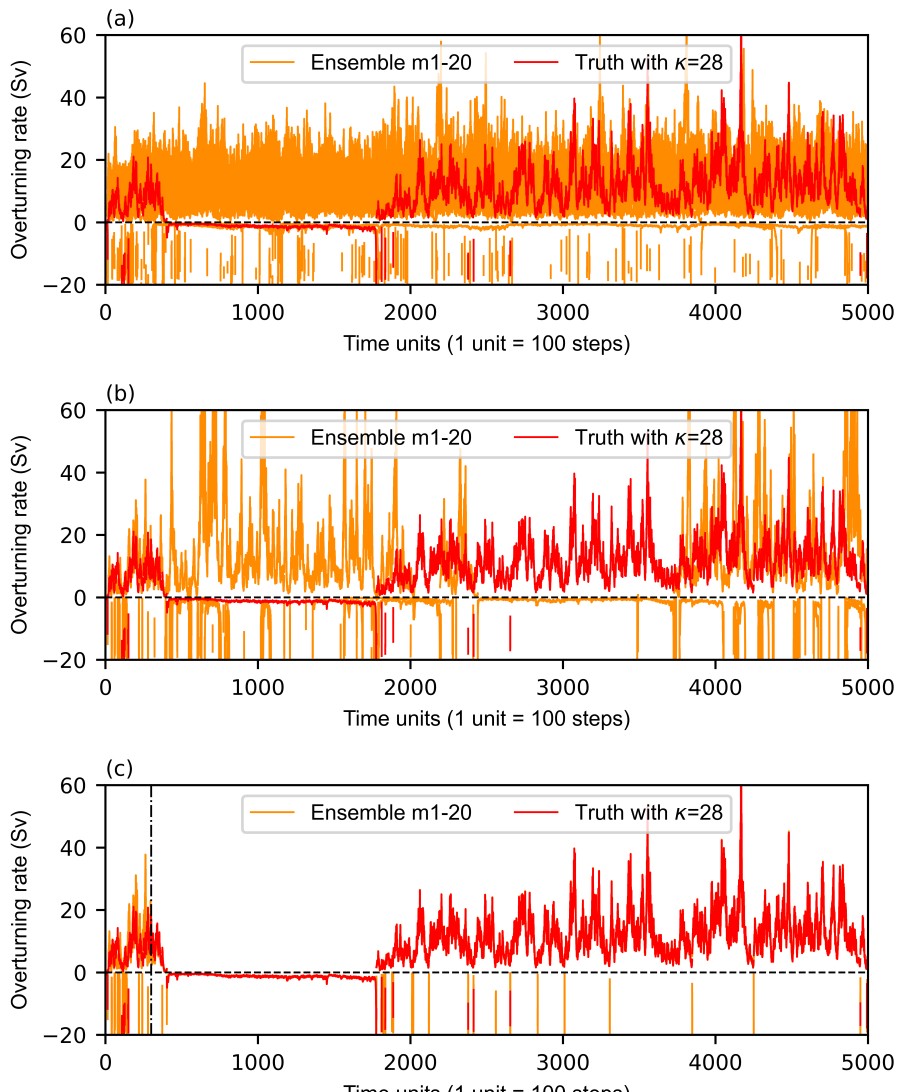

**Figure 5.** Time series of the overturning rate in the "truth" simulation with $\kappa$ = 28 (red) and the individual ensemble members (orange) in the free model control ensemble simulations a) without data assimilation and parameter estimation, b) with data assimilation or c) with data assimilation and parameter estimation using erroneously-guessed $\kappa$ values with a Gaussian perturbation that has a mean value of 32 and a standard deviation of 0.1. The dashed black line denoting $v$ = 0 is a division line between the thermal mode equilibrium state and the haline mode equilibrium state. The dotted-dashed black line denoting 300th-unit marks the start of parameter estimation.

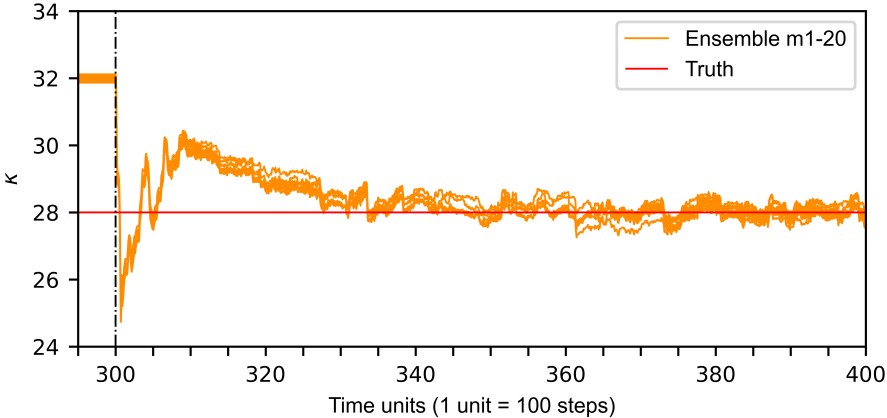

Figure 6. Time series of the estimated $\kappa$ values with a Gaussian perturbation that has a mean value of 32 and a standard deviation of 0.1 in the individual ensemble members (orange) in the free model control ensemble simulations with data assimilation and parameter estimation. The solid red line denoting $\kappa = 28$ marks the true value of $\kappa$ being estimated. The dotted-dashed black line denoting 300th-unit marks the start of parameter estimation using $x_2$, $w$, and $\eta$ observations.

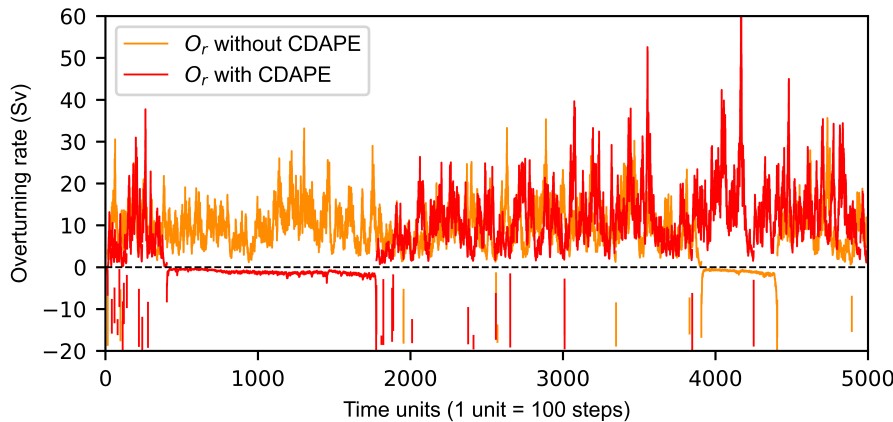

Figure 7. Time series of the overturning rate in one of the ensemble members in the free model control ensemble simulations with (red) or without (orange) data assimilation and parameter estimation, where $\kappa$ is erroneously guessed as 31.87768. The dashed black line denoting $v = 0$ is a division line between the thermal mode equilibrium state and the haline mode equilibrium state.

The 5-variable conceptual climate model could simulate the interactions between the atmosphere and ocean, and coupling it with the three-box MOC model could accurately address the main questions in this paper. The transferring of the uncertainty of the MOC3B-5V model is particularly simple and easily understood. With the help of this model, we found that the coupled model parameter estimation with observations can significantly mitigate the model deviations, thus capturing regime transitions of the AMOC. As such, the main outcome of this paper can be more readily demonstrated with this simple model. However, The MOC3B-5V model is just a simple conceptual model, and the model states $x_2$, $w$, and $\eta$ simply conceptually

simulate the variation characteristics of the atmosphere and the ocean. Although the transitions of AMOC are simulated by the MOC3B-5V model, the specific physical meaning of the model is not explicit enough. The method of capturing regime transitions in Sect. 2 is proved to be feasible in the simple model, and the next step is to apply the method to a physics-based MOC box model.

## 4 Capturing AMOC regime transitions by parameter estimation in a physics-based MOC box model

### 4.1 The MOCBM

After proving that it is feasible to capture regime transitions by constraining parameters in an idealized conceptual model, we also use a MOCBM (Tardif et al., 2014; Zhao et al., 2019) with a better physical basis to study the problem of AMOC transition. The MOCBM is a coupled lower-order ocean-atmosphere climate model constructed by Roebber (1995), which reflects the chaotic variability in the atmosphere and the oscillation or multi-equilibria in the ocean (Roebber, 1995; Taboada and Lorenzo, 540 2005). The atmospheric part of the model is represented by the wave-mean-flow atmospheric circulation model of Lorenz (1984). In contrast to Lorenz's 1963 model describing convection, Lorenz's 1984 model simulates the general atmospheric circulation at mid-latitudes (Lorenz, 1984). The ocean part of the model is represented by another three-box model of the Northern Atlantic Ocean at mid-latitude (Birchfield, 1989). A schematic illustration of this MOCBM can be found in Fig. 1 of Tardif et al. (2014).

In MOCBM, the model states (vector $x$) include atmospheric states $X$, $Y$, and $Z$, and oceanic states $T_1$, $T_2$, $T_3$, $S_1$, $S_2$, and $S_3$. The governing equations of atmosphere model are:

$$\dot{X} = -(Y^2 + Z^2) - aX + aF$$
$$\dot{Y} = XY - bXZ - Y + G$$
$$\dot{Z} = XZ + bXY - Z ,$$
(16)

where $X$ represents the intensity of the westerly wind current, $Y$ and $Z$ represent the magnitudes of cosine and sine phases of large-scale eddies, respectively. The terms $XY$ and $XZ$ represent the amplification of the eddies through interaction with the westerly current. This amplification is at the expense of the westerly current, which is denoted by the term $-(Y^2 + Z^2)$. The terms $-bXZ$ and $bXY$ represent the displacement of the eddies by westerly current, while $-aX$, $-Y$, and $-Z$ represent mechanical damping. Finally, $F$ represents the diabatic heating contrasts between the low- and high-latitude ocean, and $G$ represents the 555 longitudinal heating contrast between land and sea.

The simple non-dimensional ordinary differential equations are

$$r_1 \dot{T}_1 = \frac{1}{2} q(T_2 - T_3) + K_T(T_{A1} - T_1) - K_Z(T_1 - T_3)$$
$$r_2 \dot{T}_2 = \frac{1}{2} q(T_3 - T_1) + K_T(T_{A2} - T_2) - K_Z(T_2 - T_3)$$
$$r_3 \dot{T}_3 = \frac{1}{2} q(T_1 - T_2) + K_Z(T_1 - T_3) + K_Z(T_2 - T_3)$$

$r_1 \dot{S}_1 = \frac{1}{2} q(S_2 - S_3) - K_Z(S_1 - S_3) - Q_S$

$r_2 \dot{S}_2 = \frac{1}{2} q(S_3 - S_1) - K_Z(S_2 - S_3) + Q_S$

$r_3 \dot{S}_3 = \frac{1}{2} q(S_1 - S_2) + K_Z(S_1 - S_3) + K_Z(S_2 - S_3)$ ,      (17)

where $K_T$ represents the coefficient of heat exchange between the ocean and the atmosphere, $K_Z$ represents the coefficient of vertical interaction between the upper ocean and the deep ocean, $T_{A1}$ and $T_{A2}$ are the air surface temperature, and $Q_S$ is the

volume averaged equivalent salt flux. The non-dimensional variables $r_1$, $r_2$ and $r_3$ are defined as $r_j = V_j / (V_1 + V_2 + V_3)$, where $V_j$ represent the volume of box $j$. The meridional overturning circulation $q$ satisfies

$q = \mu[\alpha(T_2 - T_1) - \beta(S_2 - S_1)]$ ,      (18)

where $\alpha$ and $\beta$ are the thermal and haline expansion coefficients of seawater, respectively, and $\mu$ is a proportionality constant.

The coupled interaction between the ocean box model and the atmosphere model is accomplished by the terms $F$, $G$, $T_{A1}$,

$T_{A2}$, and $Q_s$. Superimposing background value and the variation in a seasonal cycle, as well as long-term variation associated with changes in upper ocean temperatures, $F$ and $G$ are defined as

$F = F_0 + F_1 \cos \omega t + F_2(T_2 - T_1)$

$G = G_0 + G_1 \cos \omega t + G_2 T_1$ ,      (19)

where $F_0$, $F_1$, $F_2$, $G_0$, $G_1$, and $G_2$ are constants, and $\omega$ is the annual frequency. Since $X$ in the atmosphere model is directly

related to the temperature, the temperature is defined as

$T_{A1} = T_{A2} - \gamma X$ ,      (20)

where $T_{A2}$ and $\gamma$ are constants. Finally, the equivalent salt flux is formulated by $Q_S = Q_{runoff} + \bar{Q}_{WV} + Q'_{WV}$ , where $Q_{runoff}$ is the runoff into the ocean from the rivers, $\bar{Q}_{WV}$ and $Q'_{WV}$ are the mean and transient eddy components of the atmospheric water vapor transport, respectively. $Q_{runoff}$ and $\bar{Q}_{WV}$ are assumed to be constant and $Q'_{WV}$ is postulated to be linearly related

to the eddy sensible heat flux ($Y^2 + Z^2$) (Stone and Yao, 1990). Finally, $Q_s$ is obtained as follows:

$Q_S = c_1 + c_2(Y^2 + Z^2)$ ,      (21)

where $c_1$ and $c_2$ are constants to be determined.

The parameters in this MOCBM are set to ($a$, $b$, $r_1$, $r_2$, $r_3$, $K_T$, $K_z$, $T_{A2}$, $\alpha$, $\beta$, $\mu$, $F_0$, $F_1$, $F_2$, $G_0$, $G_1$, $G_2$, $\gamma$,) = (0.25, 4.00, 16.495, 5.295, 1.332, 0.35, 0.05276, 1, $9.622 \times 10^{-5}$ K$^{-1}$, $7.755 \times 10^{-4}$ psu$^{-1}$, $4 \times 10^{10}$ m$^3$ s$^{-1}$, 6.65, 2.0, 47.9, -3.60, 1.24, 3.81, 0.06364).

As in Roebber (1995), the value of $Q_s$ affects the solution of the thermohaline circulation, and $Q_s$ above a critical value will eventually lead to a complete reversal of the flow. To obtain this critical value of $Q_s$, the equilibrium solution for the thermohaline circulation is calculated as the value of $Q_s$ varies from $0.5 \times 10^{-3}$ to $4.0 \times 10^{-3}$. As shown in Fig. 8, this critical value is near $2.05 \times 10^{-3}$. Considering the mean and the range of variation of ($Y^2 + Z^2$), and also referring to the values taken in Roebber (1995) and Tardif et al. (2014), $c_1$ and $c_2$ are set here to $1.94 \times 10^{-3}$ and $4.05 \times 10^{-5}$, respectively.

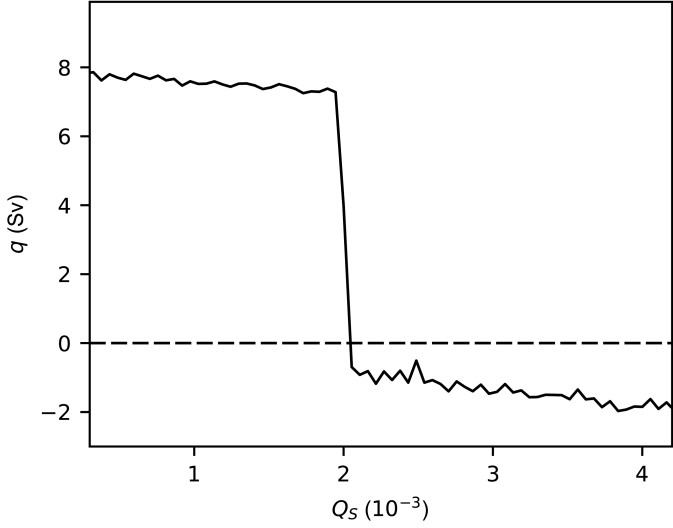

**Figure 8.** Variation of meridional overturning circulation $q$ in the space of salt flux $Q_s$. The dashed black line denoting $q = 0$ Sv is a division line between two equilibrium states.

## 4.2 Parameter estimation with observational information

A twin experimental framework is designed to perform the study of capturing regime transitions using the MOCBM. Using a fourth-order Runge–Kutta time differencing scheme with a time step of 3 h, the MOCBM is specified with parameter values described above. The truth model is spun up for an initial $10^5$ years starting from an initial value of $q$ equal to 15 Sv. Then, another 60000 years are run forward to produce the "truth" states. The states of the atmosphere and the temperature and salinity of the surface ocean are considered as the variables to be observed. The white noise is added to the "truth" states, and the standard deviations of the observation errors are set to 0.1 for $X$, $Y$, $Z$, 0.5 K for $T_1$, $T_2$, and 0.1 psu for $S_1$, $S_2$. The "observations" are eventually obtained by sampling these variables at a frequency of one year. In the twin experimental framework, the assimilation model is similar to the truth model except that parameter $\gamma$ in the box model is assumed to be incorrectly estimated, with error that is 10% greater than the standard value 0.06364. Thus, the mean value of all parameters from the twenty assimilation models is 0.070004 and their standard deviation is 10% of the standard value. The parameters in the atmosphere model or in the ocean model could be selected for parameter estimation to address the points in this paper. Given that we have experimented with the parameter in the atmosphere model before, here we show the experiment with the parameter of the ocean model. Again, the parameter being estimated is based on the model sensitivities regarding all parameters in the box model (Zhao et al., 2019).

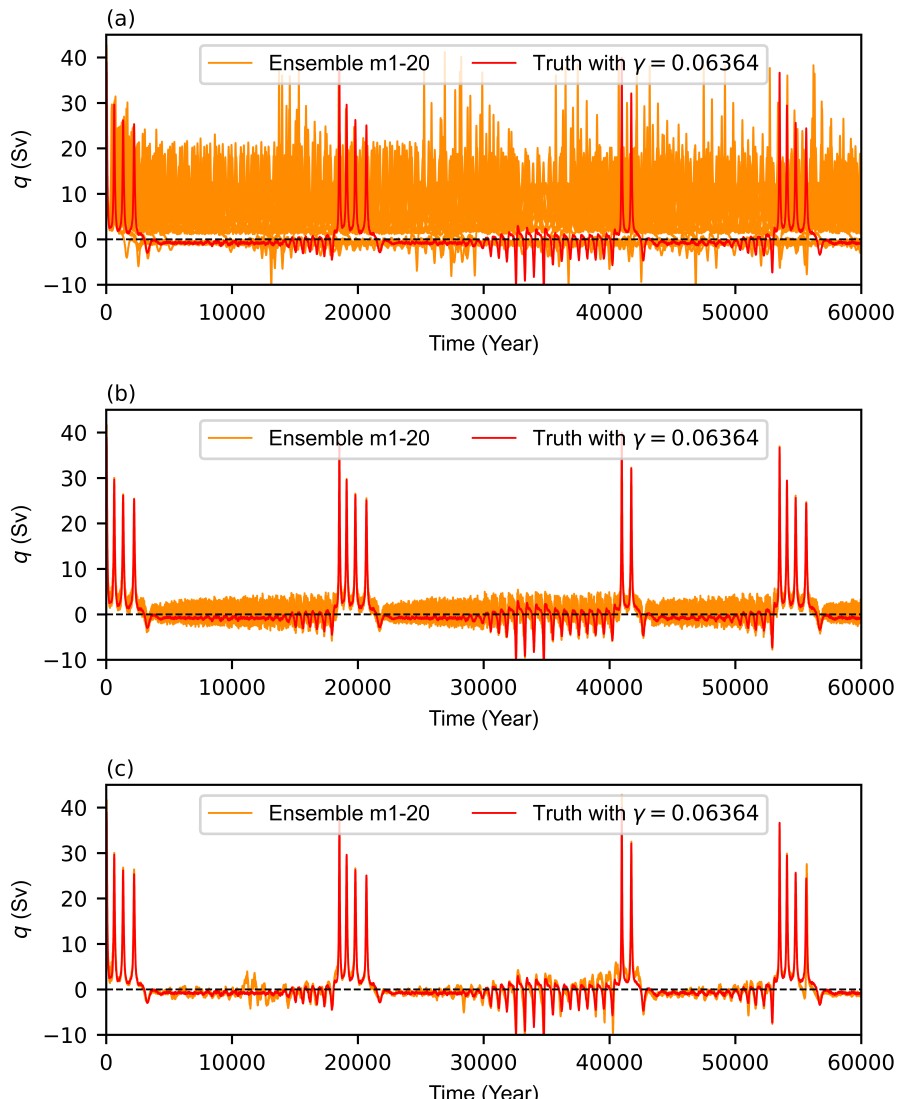

**Figure 9.** Time series of the meridional overturning circulation $q$ in the "truth" simulation with $\gamma = 0.06364$ (red) and the individual ensemble members (orange) in the free model control ensemble simulations a) without data assimilation and parameter estimation, b) with data assimilation or c) with data assimilation and parameter estimation using erroneously-guessed $\gamma$ values with a Gaussian perturbation that has a mean value of 0.070004 and a standard deviation of 10% times the standard value. The dashed black line denoting $q = 0$ is a division line between two equilibrium states.

Figure 9 shows the time series of meridional overturning circulation $q$, where the positive and negative aspects of $q$ reflect the reversed circulation, which represents the transition between two different states. The simulation results of the twenty assimilation models (orange line in Fig. 9a) are different, and they all have significant errors with the results from the truth model (red line). Then, the "observations" from the truth model are used to adjust the model states (Fig. 9b), as well as to further constrain the parameter $\gamma$ (Fig. 9c). Figure 9b shows the time series of the MOC value in the "truth" simulation (red

lines) and in the free model control ensemble simulations with only data assimilation. Due to the existence of parameter error, inaccurate analyses are obtained when only data assimilation was performed without parameter estimation. The results of data assimilation and parameter estimation are shown in Fig. 9c, where the simulation results of the twenty assimilation models (orange lines) are constrained by "observations", and the more accurate reconstructed transition path (red lines) is obtained. Since the behavior of the MOCBM and MOC3B-5V models are very similar, the figures corresponding to Fig. 6 and Fig. 7 are not shown here.

The box model in the MOCBM is based on the classical approach of adopting a buoyancy constraint, and the circulation is regulated by the surface buoyancy difference, implying that surface thermohaline forcing drives AMOC (Birchfield, 1989). In contrast, for the box model in the MOC3B-5V model, the constraint is based on mechanical energy sustaining diapycnal mixing, and the circulation is maintained by mechanical energy from wind stress and tides (Shen et al., 2011). The AMOC is driven differently in the two box models. Compared with the buoyancy constraint (MOCBM), the energy constraint (MOC3B-5V) can offer a significant advantage on rational interpretations of the transitions between thermal and haline modes (Guan and Huang, 2008). Compared to the MOC3B-5V model, the MOCBM is more explicit in physical meaning, which is mainly reflected in the meaning of the model states. The meaning of the state variables in the atmospheric part of the MOCBM is more explicit, such as $X$ for westerly wind current and $Y$ and $Z$ for large-scale eddies. The MOC3B-5V model, however, only describes the chaotic characteristic of the atmosphere starting from a simple heating disturbance problem. It further describes the basic variation characteristic of the ocean by the coupled interaction between the atmosphere and the ocean. Besides, the coupling of ocean and atmosphere in the MOCBM is sufficiently accomplished by several variables such as $F$, $T_{A1}$, and $Q_s$. The MOC3B-5V model and the MOCBM, although both are simple models, can reveal the characteristic of AMOC multi-equilibria and thus can be used to test the feasibility of the methodology in Sect. 2. By constraining the model parameters with observations, both models result in capturing regime transitions of the AMOC.

## 5 Summary and discussions

A method for combining general AMOC simulation model with ensemble Kalman filtering is designed to form a CDAPE system. Given that the discrepancy exists between the influencing factors of AMOC in the real world and the corresponding parameters of models, parameter estimation is used to estimate the model parameters. Using the CDAPE system, within a "twin" experiment framework, twenty assimilation models are set with an incorrectly estimated parameter, while a model representing the "truth" that uses the parameters as the standard values. The assimilation models simulate twenty different transition paths between AMOC states with disturbing parameters. The observational information from the "truth" is assimilated into the assimilation models, and the transition path of AMOC is optimized by parameter estimation, so that regime transitions of AMOC are captured correctly. Our results suggest that guided by estimation theory, appropriately constraining coupled model parameters with observed data can make a climate model capture regime transitions of the AMOC. The research methodology is applied to simple climate models that can simulate AMOC multi-equilibria. The first model in this study

provides conceptual proof that the methodology is feasible, and the second model with more explicit physical meaning provides further demonstration through simulation results.

A simple model that consists of a 3-box ocean model and a 5-variable climate model (the MOC3B-5V model) has been developed to simulate the basic characteristic of AMOC that transits between different equilibrium states. The parameters in the three-box model are linked to the atmospheric variables, the upper ocean variable, and the deep ocean variable in the 5-variable model to construct energy constraint, wind-driven circulation, and freshwater flux, which dynamically change within a reasonable range. By projecting observational information of model states to the parameters, the AMOC regime transitions simulated by the model are much closer to reality. It has to be noted that after the change in the three-box model from the stable haline mode to the stable thermal mode, a catastrophic change occurs in the system, which results in the disappearance of the stable haline mode (Guan and Huang, 2008). It is impossible to change the state from the thermal mode to the haline mode by changing parameters. Therefore, to adjust the transition between the thermal mode and the haline mode, additional atmospheric forcing (additional freshwater flux from the atmosphere) is added to the two boxes that have contact with the atmosphere. The effect of this forcing is small and does not affect the overall balance of the model. Although we acknowledge that the effect of additional forcing still needs to be further investigated, the disappearance of the haline mode is not addressed in this study so that we may focus on capturing regime transitions by observation-constrained model parameters. It is important to emphasize that the simple conceptual model is not attempting to simulate a specific oceanic and atmospheric physical process, but rather the opposite: our objective is to explore whether the error between models and reality in terms of the AMOC transition can be reduced by incorporating observational information into the model parameters.

The MOCBM (Roebber, 1995) with clearer physical meaning is used in this study. Since the circulation is driven only by the meridional gradients of the upper ocean temperature and salinity in the buoyancy-constraint MOCBM model, AMOC regime transitions can be captured to some extent when the upper ocean temperature and salinity are directly adjusted by data assimilation only, but the simulation results are not accurate enough. In this simple model, since the data assimilation has worked well, the contribution of parameter estimation is relatively small but still indispensable. The AMOC regime transitions are captured more accurately by parameter estimation. The degree of contributions of data assimilation or parameter estimation to the optimization of simulation results is different in these two models. Compared with the MOCBM model, the energy-constraint MOC3B-5V model is more representative for the role of parameter estimation because the circulation is maintained by mechanical energy. When leaving out the parameter estimation steps and constraining the model states only by data assimilation, the accuracy of state estimation is not high due to the existence of parameter errors. Given the fact that the circulation is driven in a more complex way in the real world, this simple model study only provides a conceptual understanding and guideline for more complex real systems such as Coupled General Circulation Model (CGCM). Although both the MOCBM and the MOC3B-5V model are simple idealized hemispheric models, our concerns can be illustrated more clearly through them. Our effort here is to make the AMOC multiple equilibrium states from model simulations better reflect the features in reality. Our focus is on adjusting the model parameters by sampling the observations so that the simulation of the model is closer to the truth on the feature of regime transitions. The conceptual model, albeit simple, has demonstrated the

importance of data assimilation and parameter estimation. It is hoped that such a simple model study on AMOC transition will inspire the hypotheses and the optimization of parameters in CGCMs. Taking a study by Ashkenazy and Tziperman (2007) as an example, to understand the ocean general circulation model results, they constructed a simple three-box model to understand the behavior of the thermohaline circulation in more realistic parameter regimes (Ashkenazy and Tziperman, 2007).

    We have already captured regime transitions of the AMOC in a conceptual model as well as in a simple model with clearer

physical meaning and will apply this method to more complex real systems such as CGCMs. The characteristic of AMOC multi-equilibria has been simulated in box models (e.g., Stommel, 1961; Rooth, 1982; Welander, 1986; Birchfield, 1989), ocean circulation models (e.g., Marotzke and Willebrand, 1991), and coupled ocean-atmosphere models (e.g., Manabe and Stouffer, 1988). However, it should also be noted that AMOC multiple equilibria have not been directly simulated by some CGCMs. Tremendous research efforts thus have been put to tackle this issue. One focus was on the CGCM presentation of

Stommel's salt advection feedback (Rahmstorf, 1996). It has been suggested this feedback is distorted in CGCMs due to salinity biases (Huisman et al., 2010; Jackson, 2013; Liu et al., 2017). Another argument is on ocean eddies. It has also been suggested that CGCMs with an eddy-permitting ocean allow for a simulation of AMOC multiple equilibria (Jackson and Wood, 2018) since ocean eddies modify the overall freshwater balance (Mecking et al., 2016). In follow-up studies, we will explore the contribution of a CDAPE system to AMOC multi-equilibrium using different resolutions, ranging from a coarse-resolution

CGCM with the ability to simulate AMOC multi-equilibrium characteristic, and eventually to a high-resolution and more realistic Earth system model. In a recent study, two types of AMOC transitions were described, with a temporary cessation of the downwelling (called F-type transition) or a full collapse of the AMOC (called S-type transition), and the F-type transitions might have been found in the direct observation (Castellana et al., 2019, 2020). The general methodology of this study could be used for both S-type transition and F-type transition. The S-type transition with centurial and millennial timescales could

use observations from paleoclimate records, and the methodology can be applied to paleoclimate models for capturing AMOC regime transitions. In current climate system, the F-type transition with very high transition probabilities on multi-decadal timescales (Castellana et al., 2019; Castellana and Dijkstra, 2020) could use direct observations from RAPID or indirect observations from satellites or ARGO program.

    Although the observation-constrained model simulates the transition between different equilibrium states of AMOC, this

study only serves as the first step of capturing regime transitions and many challenges still exist. First, the deviations of AMOC transition paths simulated in different models are caused by not only parameter errors but also mismatches between real physical processes and model simulations (Zhang et al., 2012). Therefore, the performance of parameter estimation still needs further experimentation with more realistic models. Second, the mechanism of AMOC transition needs further investigation. The effect of stochastic forcing has been taken into account in previous work. Cessi (1994) studied the transition from one

equilibrium to another in a modified Stommel model and she found that the transition could occur under stochastic white-noise forcing. In our study, the transition phenomenon of AMOC is ultimately affected by the model parameters. Usually, traditional state estimation with data assimilation has limited usage to detect the mechanism. Here, we aim at constraining the model parameters through utilization of observational information, which eventually results in a more realistic model behavior in

terms of the AMOC transitions. A future effort is needed on how the effect of the stochastic component will manifest in the AMOC system. Third, Aksoy et al. (2006b) proposed a spatial updating technique that recovers the globally uniform parameter value using a spatial average of the entire spatially varying parameter field. Wu et al. (2012, 2013) explored the impact of the geographic dependence of observing system on the parameters. The adjustment of the parameters is based on the spatial distribution of the model state sensitivity to parameters. Liu et al. (2014a, b) proposed the adaptive spatial average method that obtains the final global uniform posterior parameter based on spatially varying posterior estimated parameter values. In this study, considering that the simple box models are used as a first step to explore AMOC transitions, it is more appropriate to use the identity model. The impact of geographic-dependent parameter optimization on climate estimation and prediction can be considered in future studies for complex systems such as CGCMs.

Besides, in the study of two simple models, the observational information, which is used for data assimilation and parameter estimation, only comes from the atmosphere and surface ocean. In the real Earth system, the flow of seawater located in the deep ocean is an important part of AMOC, its measurement is difficult. The changes in each component of AMOC will affect the entire circulation. AMOC reconstruction heavily relies on comprehensive observational data. In the future, with the improvement of the Earth observing system, the coupled climate system model will be improved continuously, and the results of numerical simulation will have a higher credibility. These could lead to significant improvement of the reanalysis and prediction of the AMOC.

## Code and data availability

All corresponding codes and simulation data in this work are available from the corresponding author by sending a request (szhang@ouc.edu.cn).

## Author contribution

All the authors designed the study. ZL implemented the study with guidance from SZ, YG, YS, and XD. ZL prepared the manuscript, and all authors jointly edited and revised it.

## Competing interests

The authors declare that they have no conflict of interest.

## Acknowledgements

This research is supported by the National Key R&D Program of China (grant nos. 2017YFC1404100 and 2017YFC1404104), the National Natural Science Foundation of China (grant nos. 41775100, 41830964), the Shandong Province's "Taishan" Scientist Project (grant no. ts201712017), and the Qingdao City "Creative Leadership" Program (grant no. 19-3-2-7-zhc).

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
