# Peer review of "A Study of Capturing AMOC Regime Transition through Observation-Constrained Model Parameters"

_Nonlinear Processes in Geophysics, 2021_

## Author Comment (AC1)

**Response to Reviewer #1**

**Comments from Reviewer #1:**

**In this manuscript the authors describe a series of twin experiments in which the Ensemble Adjustment Kalman Filter (EAKF) is used for state and parameter estimation in models of the Atlantic Meridional Overturning Circulation (AMOC). The models in question are box models, driven by atmospheric input taken from the output of chaotic, rather than stochastic systems. The authors demonstrate that their methods allow reliable estimation of unknown parameters in the model AMOCs, which exhibit multiple stable equilibria. The results described here will be of interest to a broad segment of NPG readers, but the manuscript as it stands needs a great deal of work to make it acceptable.**

RE: A few conferences have been held for all co-authors to discuss the comments from reviewer #1. All authors converge to the point that the constructive comments and suggestions are extremely thoughtful and important for improving the manuscript and enhancing our understanding on the topic. Several extra experiments for addressing the concerns of the reviewer are performed. Thanks for your encouragement. All issues are replied point-by-point as below. We hope the whole manuscript has been essentially improved.

**Application of data assimilation methods to use observations to track the evolution of solutions of systems that exhibit multiple stable modes has been described by many authors since the 1990s; see, e.g., Weir et al., Nonlin Proc Geophys 2013 and references therein. A look through the literature since the 1990s will also turn up examples of simultaneous state and parameter estimation in simple systems. The novelty of the present work lies in the specific application to regime transitions in the AMOC. I don't know of other examples of application of optimized methods such as the EAKF to joint state and parameter estimation of box models of the AMOC, which have been around since Stommel (1961).**

RE: Excellent suggestion! We have added more detailed statements about the application of data assimilation methods (L112-118) and the examples of simultaneous state and parameter estimation in nonlinear systems having multiple stable modes (L122-128).

L112-118:
Tardif et al. (2014) implement data assimilation with EnKF to recover the AMOC with observations in a low-order coupled atmosphere-ocean climate model. They mainly explore the value of data assimilation for the initialization of the AMOC, while the effect of parameter errors in AMOC simulations needs further discussion. As another class of ensemble-based assimilation methods, particle filters, unlike the EnKF, are applicable to non-Gaussian probability distributions (e.g., Gordon et al., 1993; van Leeuwen, 2009). A mixture-based implicit particle method is presented and could detect transitions in an example with multiple attracting states (Weir et al., 2013a). However, the particle filter is plagued by the curse of dimensionality as the system dimension increases (Snyder et al., 2008; Carrassi et al., 2018).

L122-128:

Based on EAKF, a data assimilation scheme for enhanvce parameter correction is designed to improve parameter estimation using observations (Zhang et al., 2012). Zhao et al. (2019) perform this scheme in a simple AMOC box model, and the model parameters are successfully optimized when the model errors are caused by only erroneously set parameters. Although the AMOC regime transition is not addressed in their study, their exploration of model sensitivities regarding parameters serves as a guideline for our research. Many efforts have been made to advance the application of data assimilation and parameter estimation in nonlinear systems having multiple equilibrium states (e.g., Miller et al., 1994, 1999; Khalil et al., 2009; Weir et al., 2013b; Bisaillon et al., 2015).

**The authors need to be more specific about exactly how their results differ from existing results, and why they are interesting. The closest thing that I can find to a statement of purpose appears at the end of the introduction, p4, beginning on line 111: "Here we present a method for improving the modeling of AMOC multi-equilibria. The new method is shown to simulate the AMOC transition between different equilibrium states accurately in two simple coupled models …" As noted above, others have shown the ability to simulate transition between different equilibrium states in other systems. If the authors' methods are novel, they should point out their differences from other methods that have been applied to similar systems.**

RE: Thank you for this suggestion! In this revision, we have reorganized the existing literature on AMOC transitions (L81-90), as well as more specifically illustrated how this study differs from previous studies and pointed out the novelty of this paper (L96-103, L128-133).

L81-90:

AMOC transitions can occur due to external forcing or internal feedback (Klockmann et al., 2020). The external forcing applied in systems may include freshwater forcing (e.g., Cessi, 1994; Castellana et al., 2019), wind forcing (e.g., Ashkenazy and Tziperman, 2007; Kleppin et al., 2015), ice sheet forcing (e.g., Zhang et al., 2014; Mitsui and Crucifix, 2017), $CO_2$ forcing (e.g., Zhang et al., 2017). The physical processes in the model are changed by external forcing, resulting in the transition between different states of the AMOC. For the AMOC model without external forcing, the transition is triggered by complex internal interactions within the model, such as salt oscillations (Peltier and Vettoretti, 2014), internal oceanic processes (Sévellec and Fedorov, 2014), thermohaline oscillations (Brown and Galbraith, 2016), intermittencies in the sea-ice cover (Gottwald, 2021). Regardless of whether it is due to external forcing or internal feedback, AMOC transitions could be influenced by complex physical processes in models, and the parameters involved in these physical processes are usually fixed.

L96-103:

Observation-constrained model parameters no longer keep fixed values but are constantly varying over time. The purpose of this paper is to explore whether the variations of observation-constrained parameters that allow the physical processes of model to evolve over time can make the simulation results closer to the "observed" feature of regime transitions. The models in this paper are obtained by coupling AMOC box model with Lorenz's model, similar to the work by Roebber (1995) or Gottwald

(2021), where the variation of AMOC is driven by the chaotic dynamical system. The thermal mode and the reverse haline mode correspond to different equilibrium states of the AMOC. For simplicity, we will refer to these different states as the stronger AMOC (on-state) and the weaker AMOC (off-state) in simple conceptual models (e.g., Weijer et al., 2019).

L128-133:
Although the AMOC model eventually exhibits multiple equilibria, the AMOC is not a direct model state but is indirectly derived from model states such as atmospheric wind, ocean temperature and salinity. Instead of adjusting AMOC directly, the model states are adjusted through data assimilation. When constraining model parameters by observational information, the parameters that constantly vary with observations may provide more diversity in the physical processes involved with AMOC regime transition, so that the model can simulate more AMOC transition paths.

**Their presentation of their three box model, equations (5)-(10) is confusing. Why different systems of equations for the thermal and saline modes? For details of the model they refer the reader to Shen et al. (2011) and the model described there is a single system that exhibits both saline and thermal modes. The model they finally use, defined in equation (12), is a complex system with many switches. I don't understand why this is necessary. Why not just use some form of (5) from Shen et al. (2011)?**

RE: Sorry we haven't explained this clearly. Shen et al. (2011) described two different systems of equations for the thermal and haline modes, which are exactly the same as the equations (5)-(10) in this paper. They do not show the equations for the haline mode, but describe the equations for the thermal mode in detail. The haline mode is described briefly in a sentence between equation (2d) and equation (3) in Shen et al. (2011). Following this suggestion, we have added new lines to clarify this issue in L301-303, L318-320, and L401-402.

L301-303:
The governing equations for the haline mode also follow the study of Shen et al. (2011). They didn't show those equations, but only described them briefly. In this paper, to describe the construction of MOC3B-5V more clearly later, those equations are shown here.

L318-320:
Similar equations for the thermal and haline modes could be found in Guan and Huang (2008) for Eq. (1) (thermal mode) and Eq. (2) (haline mode), and in Shen and Guan (2015) for Eqs. (1)-(6) (thermal mode) and Eqs. (7)-(9) (haline mode).

L401-402:
A similar AMOC box model with many switches could be found in Castellana et al., (2019).

**The results they get from this model, driven by a chaotic atmosphere, are encouraging. Figures 5-7 show that the data assimilation/parameter estimation system works well, reproducing the "true" trajectory quite accurately and producing a good estimate of the unknown parameter, while leaving out the parameter estimation steps, and using EAKF for state estimation without adjusting the parameter to its true value does not do nearly so well.**

RE: Thanks for your encouragement.

**At the end of this section the authors state: "The MOC3B-5V model is just a simple conceptual model, and the model states x2, w, and η simply conceptually simulate the variation characteristics of the atmosphere and the ocean. Although the transitions of AMOC are simulated by the MOC3B-5V model, the specific physical meaning of the model is not explicit enough. The method of capturing regime transitions in Sect. 2 is proved to be feasible in the simple model, and the next step is to apply the method to a physics-based MOC box model." The next section describes experiments with a "physics-based MOC box model," which is no more complicated than the MOC3B-5V model in the previous section, and the authors do not make clear what conceptual points are made with the MOC3B-5V model that are any less clear in the "MOCBM." It seems to me that the sections dealing with the MOC3B-5V model, i.e., much of section 2 and all of section 3 could be left out entirely without any loss of understanding on the part of the reader, however convincing figures 5-7 may be. It shouldn't be too hard for the authors to include additional figures corresponding to figures 5b, 6 and 7 to illustrate the details of the MOCBM experiment**

RE: Thanks for your thoughtful suggestion! In this revision, we address this comment by two parts. In the first part, we like to show the details of the MOCBM experiment (additional Figures S1, S2, and S3 below corresponding to Figures 5, 6 and 7 in the manuscript). Based on your suggestion, we have added Fig. S1b that shows the assimilation result to the revised manuscript (Fig. 9b) and more relevant discussions have been added in L613-615. Given that Figs. S2 and S3 are very similar to the behavior of the MOC3B-5V model, they are not shown in the manuscript. However, the corresponding descriptions have been added in L618-619.

In the second part, we have added more discussions to justify the necessity of MOC3B-5V and MOCBM models as three aspects. First, new lines have been added to emphasize the benefits of the MOC3B-5V model in L520-524. Second, more discussions about the important differences between these two models have been added in L620-623. Third, we have added more description and discussion about the different behaviors of different models in data assimilation and parameter estimation to optimize the simulation results in L664-675.

L613-615:
Figure 9b shows the time series of the MOC value in the "truth" simulation (red lines) and in the free model control ensemble simulations with only data assimilation. Due to the existence of parameter error, inaccurate analyses are obtained when only data assimilation was performed without parameter estimation.

L618-619:

Since the behavior of the MOCBM and MOC3B-5V models are very similar, the figures corresponding to Fig. 6 and Fig. 7 are not shown here.

L520-524:

The 5-variable conceptual climate model could simulate the interactions between the atmosphere and ocean, and coupling it with the three-box MOC model could accurately address the ideas in this paper. The transferring of the uncertainty of the MOC3B-5V model is particularly simple and easily understood. With the help of this model, we found that the coupled model parameter estimation with observations can significantly mitigate the model deviations, thus capturing regime transitions of the AMOC. As such, this simple model has more visibility to demonstrate the essence of the problem.

L620-623:

The box model in the MOCBM is based on the classical approach of adopting a buoyancy constraint, and the circulation is regulated by the surface buoyancy difference, implying that surface thermohaline forcing drives AMOC (Birchfield, 1989). In contrast, for the box model in the MOC3B-5V model, the constraint is based on mechanical energy sustaining diapycnal mixing, and the circulation is maintained by mechanical energy from wind stress and tides (Shen et al., 2011).

L664-675:

Since the circulation is driven only by the meridional gradients of the upper ocean temperature and salinity in the buoyancy-constraint MOCBM model, AMOC regime transitions can be captured to some extent when the upper ocean temperature and salinity are directly adjusted by data assimilation only, but the simulation results are not accurate enough. In this simple model, since the data assimilation has worked well, the contribution of parameter estimation is relatively small but still indispensable. The AMOC regime transitions are captured more accurately by parameter estimation. The degree of contributions of data assimilation or parameter estimation to the optimization of simulation results is different in these two models. Compared with the MOCBM model, the energy-constraint MOC3B-5V model is more representative for the role of parameter estimation because the circulation is maintained by mechanical energy. When leaving out the parameter estimation steps and constraining the model states only by data assimilation, the accuracy of state estimation is not high due to the existence of parameter errors. Given the fact that the circulation is driven in a more complex way in the real world, this simple model study only provides a conceptual understanding and guideline for more complex real systems such as Coupled General Circulation Model (CGCM).

[Figure]

**Figure S1**. Time series of the meridional overturning circulation $q$ in the "truth" simulation with $\gamma = 0.06364$ (red) and the individual ensemble members (orange) in the free model control ensemble simulations a) without data assimilation and parameter estimation, b) with data assimilation or c) with data assimilation and parameter estimation using erroneously-guessed $\gamma$ values with a Gaussian perturbation that has a mean value of 0.070004 and a standard deviation of 10% times the standard value. The dashed black line denoting $q = 0$ is a division line between two equilibrium states.

[Figure]

**Figure S2**. Time series of the estimated $\gamma$ values with a Gaussian perturbation that has a mean value of 0.070004 and a standard deviation of 10% times the standard value in the individual ensemble members (orange) in the free model control ensemble simulations with data assimilation and parameter estimation. The solid red line denoting $\gamma = 0.06364$ marks the true value of $\gamma$ being estimated. The dotted-dashed black line denoting 800th-year marks the start of parameter estimation using the observations of the atmosphere states and the temperature and salinity of the surface ocean.

[Figure]

**Figure S3**. Time series of the meridional overturning circulation $q$ in one of the ensemble members in the free model control ensemble simulations with (red) or without (orange) data assimilation and parameter estimation, where $\gamma$ is erroneously guessed as 0.067419. The dashed black line denoting $q = 0$ is a division line between the thermal mode equilibrium state and the haline mode equilibrium state.

**The two systems dealt with here are highly parameterized, but the parameter the authors chose to estimate, in both cases, was a parameter in the atmospheric model, equations (11) and (16). Why didn't they choose a parameter in the box models?**

RE: Thanks for your suggestion. We have re-chosen the parameter to be estimated in the second system. The parameter $b$ in the atmospheric model is replaced by the parameter $\gamma$ in the box model. Following this suggestion, the first paragraph in Section 4.2 has been rewritten (L595-598), and Fig. 9 has been replaced by a new figure (L603-608). The new results are similar to the original results for parameter $b$. Both can demonstrate that the AMOC regime transitions can be captured by constraining the model parameters with observations.

Besides, the discussion about parameter selection has been added in L598-602. Considering the length limitation, the process of investigating the model sensitivity is not shown in the manuscript. However, the relevant literature has been added in L601-602. The process of investigating the model sensitivity will be described later, and we first show the results in Figure S4. The most sensitive parameter $T_{A2}$ directly affects the temperature in the box model, so it is more appropriate to choose the parameter $\gamma$ in comparison.

L595-598:
In the twin experimental framework, the assimilation model is similar to the truth model except that parameter γ in the box model is assumed to be incorrectly estimated, with error that is 10% greater than the standard value 0.06364. Thus, the mean value of all parameters from the twenty assimilation models is 0.070004 and their standard deviation is 10% of the standard value.

L603-608: (The same as Fig. S1).

L598-602:
The parameters in the atmosphere model or in the ocean model could be selected for parameter estimation to address the points in this paper. Given that we have experimented with the parameter in the atmosphere model before, here we show the experiment with the parameter of the ocean model. Again, the parameter being estimated is based on the model sensitivities regarding all parameters in the box model (Zhao et al., 2019).

L601-602:
Again, the parameter being estimated is based on the model sensitivities regarding all parameters in the box model (Zhao et al., 2019).

[Figure]

**Figure S4**. The sensitivity percentage of MOC in the space of the parameters. The sensitivity percentage refers to the ratio of the specified parameter sensitivity to the total sum of the sensitivities for the 12 parameters in the box model. The time-averaged value of the MOC spread is calculated by data over the last 200 years.

Referring to Section 3.2.3 in Zhao et al. (2019), the process of investigating the model sensitivity regarding each parameter is as follows. Each parameter is individually perturbed into 20 members from the ensemble as white noise, with a standard deviation of 10% of its standard value, while the remaining 11 parameters remain fixed at their standard values. Then, from the same initial states, all 20 ensemble members are freely run for 250 years. Only the model outputs of the last 200 years are used to quantitatively calculate the relative sensitivities (Owing to the MOC value is a long-time-scale variable). We use the standard deviation in the MOC strength to evaluate the model sensitivity regarding each parameter. The time-averaged sensitivity percentages of the MOC strength with respect to all parameters are shown in Fig. S4.

---

## Author Comment (AC2)

**Response to Reviewer #2**

**Comments from Reviewer #2:**

**General Comments:**

This paper addresses regime transition in AMOC though state and parameter estimation, via application of the EAKF. While the idea of parameter estimation is not new in ensemble-based data assimilation and there are many published papers addressing it, I believe that the novelty of the paper is the application of EAKF to regime transition in AMOC. Still, the paper needs to explain what the main findings are and how those findings could improve our knowledge about AMOC. In addition, I have some specific comments requiring clarification of the parameter estimation approach and the EAKF equations.

RE: A few conferences of all co-authors have been held to discuss the comments of reviewer #2. All authors appreciate greatly for the encouragements and comments. All the comments are very important and useful for authors to improve the quality of this manuscript. The paper is renewed as the reviewer's suggestions. Thanks for your encouragement. In this revision, a more detailed explanation has been added to the Introduction section. We have expressed better what the main findings of this paper are (L96-103, L664-675) and how these findings could improve our knowledge about AMOC (L128-133). All specific comments are replied point-by-point as below. We hope the whole manuscript has been essentially improved.

**L96-103:**

Observation-constrained model parameters no longer keep fixed values but are constantly varying over time. The purpose of this paper is to explore whether the variations of observation-constrained parameters that allow the physical processes of model to evolve over time can make the simulation results closer to the "observed" feature of regime transitions. The models in this paper are obtained by coupling AMOC box model with Lorenz's model, similar to the work by Roebber (1995) or Gottwald (2021), where the variation of AMOC is driven by the chaotic dynamical system. The thermal mode and the reverse haline mode correspond to different equilibrium states of the AMOC. For simplicity, we will refer to these different states as the stronger AMOC (on-state) and the weaker AMOC (off-state) in simple conceptual models (e.g., Weijer et al., 2019).

**L664-675:**

Since the circulation is driven only by the meridional gradients of the upper ocean temperature and salinity in the buoyancyconstraint MOCBM model, AMOC regime transitions can be captured to some extent when the upper ocean temperature and salinity are directly adjusted by data assimilation only, but the simulation results are not accurate enough. In this simple model, since the data assimilation has worked well, the contribution of parameter estimation is relatively small but still indispensable. The AMOC regime transitions are captured more accurately by parameter estimation. The degree of contributions of data assimilation or parameter estimation to the optimization of simulation results is different in these two models. Compared with the MOCBM model, the energy-constraint MOC3B-5V model is more representative for the role of parameter estimation because the circulation is maintained by mechanical energy. When leaving out the parameter estimation steps and constraining the model states only by data assimilation, the accuracy of state estimation is not high due to the existence of parameter errors. Given the fact that the circulation is driven in a more complex way in the real world, this simple model study only provides a conceptual understanding and guideline for more complex real systems such as Coupled General Circulation Model (CGCM).

**L128-133:**

Although the AMOC model eventually exhibits multiple equilibria, the AMOC is not a direct model state but is indirectly derived from model states such as atmospheric wind, ocean temperature and salinity. Instead of adjusting AMOC directly, the model states are adjusted through data assimilation. When constraining model parameters by observational information, the parameters that constantly vary with observations may provide more diversity in the physical processes involved with AMOC regime transition, so that the model can simulate more AMOC transition paths.

**Specific Comments:**

**(1) Please explain how equations (1)-(4) were derived from the EAKF.**

RE: Good suggestion! In this revision, a detailed derivation process is shown below. Given the length limitation, these derivations are not added to the manuscript. However, a concise description and the corresponding references have been added in L192-197.

**L192-197:**

The ensemble adjustment Kalman filter (Anderson, 2001) is used for data assimilation and parameter estimation in this study. The basic process of the two-step EAKF (Anderson, 2003; Zhang and Anderson, 2003; Zhang et al., 2007) is to project the observational increment onto model states (relevant parameters) by calculating the error covariance between the prior ensemble of the model variable (parameter) and the model-estimated ensemble. The core of the two-step EAKF is to calculate the increment of each state variable by a global least squares fit (linear regression), and the calculation of the observational increment is related to the scalar application of the equations of EAKF (Anderson, 2003).

Based on the EAKF (Anderson, 2001), Anderson (2003) described a two-step data assimilation procedure for ensemble filtering under a local least squares framework.

The joint state-observation space is defined by the joint space state vector:  $\mathbf{z} = [\mathbf{x}, \mathbf{y}]$ , where  $\mathbf{x}$  is the model state vector;  $\mathbf{y} = h(\mathbf{x})$ , where h is the forward observation operator. Using Bayesian statistics, the distribution of the posterior (or updated) distribution can be computed from the prior distribution, as

$$\mathbf{p}(\mathbf{z}^u) = \mathbf{p}(\mathbf{y}^o | \mathbf{z}^p) \mathbf{p}(\mathbf{z}^p) / (\text{norm}) \quad . \tag{S1}$$

At the heart of the ensemble Kalman filter is the fact that the product of the joint prior Gaussian with mean  $\bar{z}^p$ , covariance

 $\Sigma^p$ , and the Gaussian observation distribution with mean  $\mathbf{y}^o$  and error variance  $\mathbf{R}$  has covariance

$$\boldsymbol{\Sigma}^{u} = [(\boldsymbol{\Sigma}^{p})^{-1} + \mathbf{H}^{\mathrm{T}} \mathbf{R}^{-1} \mathbf{H}]^{-1} , \qquad (S2)$$

and mean

$$\bar{\mathbf{z}}^u = \mathbf{\Sigma}^u [(\mathbf{\Sigma}^p)^{-1} \bar{\mathbf{z}}^p + \mathbf{H}^T \mathbf{R}^{-1} \mathbf{y}^o]$$
(S3)

The EAKF constructs an updated ensemble with a mean and sample variance that satisfy Eq. (S2) and Eq. (S3). In Anderson (2001), this is done by shifting the mean of the ensemble and then adjusting the spread of the ensemble around the updated mean using a linear operator **A**:

$$\mathbf{z}_{i}^{u} = \mathbf{A} \left( \mathbf{z}_{i}^{p} - \bar{\mathbf{z}}^{p} \right) + \bar{\mathbf{z}}^{u}$$
(S4)

where **A** satisfies  $\Sigma^{u} = \mathbf{A}\Sigma^{p}\mathbf{A}^{T}$ .

In everything that follows, results are presented only for assimilation of a single scalar observation. Define the joint state space forward observation operator for a single observation as the order  $1 \times k$  linear operator  $\mathbf{H} = [0,0, ...,0,1]$ , where k is the joint state space size. The updated probability for the marginal distribution of the observation joint state variable y can be formed:  $p_y(y^u) = p(y^o|y^p)p_y(y^p)/(\text{norm})$ , where the subscript on the probability densities indicates a marginal probability on the observation variable. Note that this equation does not depend on any of the model state variables. This suggests a partitioning of the assimilation of an observation into two parts. The first determines updated ensemble members for the observation variable given the observation. To update the ensemble sample of  $y^p$ , an increment is computed for each ensemble member:

$$\Delta y_i = y_i^u - y_i^p \quad , \tag{S5}$$

where i = 1, ..., N, and N is the ensemble size. The second step computes corresponding increments for *i*-th ensemble sample of *j*-th state variable  $\Delta x_{i,j}$ . This requires assumptions that the prior distribution is Gaussian. This equivalent to assuming that a least squares fit to the prior ensemble members summarizes the relationship between the joint state variables.

**Figure S1.** An idealized representation showing the relation between update increments for a state variable x and an observation variable y for a five member ensemble represented by asterisks. The projection of the ensemble on the x and y axes is represented by a plus sign and the observation  $y^o$  is represented by "\*". The gray dashed line shows a global least squares fit to the ensemble members. Update increments for ensemble members 1 and 4 for y are shown along with corresponding increments for the ensemble as a whole (thin vectors parallel to least squares fit) and for the x ensemble. From Anderson (2003)

Figure S1 depicts the simplest example in which there is only a single state variable x. The observation variable y is related to x by the operator h, which is nonlinear in the figure. Increments for each ensemble sample of y have been computed. The corresponding increments for x are then computed by a global least squares fit (linear regression) so that

$$\Delta x_i = \frac{cov(x_i, y_i)}{(\sigma^p)^2} \Delta y_i \quad , \tag{S6}$$

where  $cov(x_i, y_i)$  is the prior covariance of x with y,  $(\sigma^p)^2$  is the prior variance of y.

Equation (S1) implies that the observation variable can be updated independently of the other joint state variables. Using a scalar application of Eq. (S2), the updated variance for y can be written

$$(\sigma^{u})^{2} = [((\sigma^{p})^{2})^{-1} + ((\sigma^{o})^{2})^{-1}]^{-1} = \frac{(\sigma^{o})^{2}(\sigma^{p})^{2}}{(\sigma^{o})^{2} + (\sigma^{p})^{2}} .$$
(S7)

Applying a scalar version of Eq. (S3) to compute the updated mean

$$\bar{y}^{u} = (\sigma^{u})^{2} [((\sigma^{p})^{2})^{-1} \bar{y}^{p} + ((\sigma^{o})^{2})^{-1} y^{o}] = \frac{(\sigma^{o})^{2}}{(\sigma^{o})^{2} + (\sigma^{p})^{2}} \bar{y}^{p} + \frac{(\sigma^{p})^{2}}{(\sigma^{o})^{2} + (\sigma^{p})^{2}} y^{o} .$$
(S8)

Using a scalar application of Eq. (S4), the updated value of y can be written

$$y_i^u = \alpha \left( y_i^p - \bar{y}^p \right) + \bar{y}^u , \qquad (S9)$$

where  $\alpha = \sqrt{(\sigma^u)^2 / (\sigma^p)^2} = \sqrt{\frac{(\sigma^o)^2}{(\sigma^o)^2 + (\sigma^p)^2}} .$

Hence, the observational increment from Eq. (S5) can be written

$$\Delta y_i = y_i^u - y_i^p = \alpha \left( y_i^p - \bar{y}^p \right) + \bar{y}^u - y_i^p$$

=  $\sqrt{\frac{(\sigma^o)^2}{(\sigma^o)^2 + (\sigma^p)^2}} \left( y_i^p - \bar{y}^p \right) + \frac{(\sigma^o)^2}{(\sigma^o)^2 + (\sigma^p)^2} \bar{y}^p + \frac{(\sigma^p)^2}{(\sigma^o)^2 + (\sigma^p)^2} y^o - y_i^p$ . (S10)

For more details, please see Sect. 2c and 3b in Anderson (2003).

Also, please explain exact meaning of each variable. For example, is  $\Delta\beta^p$  a value of the parameter  $\beta$ , or prior ensemble perturbation of the parameter  $\beta$ ? Is  $\Delta y_i^p$  prior ensemble spread or prior ensemble perturbation? In addition, all vectors and matrices need to have clearly defined space (e.g., observation or state space) and dimensions (e.g., Nobs, Nstate, Nens, Nobs x Nens, ...).

RE: Excellent suggestion! To fully address this comment, we have substantially rewritten Section 2.2. The meaning of each variable has been explained more exactly, and the space and dimensions of all vectors and matrices have been clearly defined. We believe that the new Section 2.2 has been considerably improved (L197-220).

**L197-220:**

All observations at time t have the observation value  $y^o$  (in  $N_{obs}$  dimensions). For a single observation  $y^o_k$  at the k-th observation location ( $k = 1 \sim N_{obs}$ ), the standard deviation of observational error is  $\sigma^o$  (assumed to be Gaussian). The model states are mapped onto the observational space by applying a linear interpolation, and then the prior (model-estimated) ensemble of the k-th observation  $y^p_k$  (in  $N_{ens}$  dimensions) can be obtained.  $y^p_{k,i}$  is the *i*-th prior ensemble member of the k-th observation. The ensemble mean and standard deviation are  $\bar{y}^p_k$  and  $\sigma^p_k$ , respectively.

The first step is to compute the observational increment of the *k*-th observation ( $k = 1 \sim N_{obs}$ ). The observational increment  $\Delta y_{k,i}^o$  for the *i*-th ensemble member ( $i = 1 \sim N_{ens}$ ) is formulated by

$$\Delta y_{k,i}^{o} = \bar{y}_{k}^{u} + \Delta y_{k,i}^{\prime} - y_{k,i}^{p} \quad , \tag{1}$$

where  $\bar{y}_k^u$  is the posterior ensemble mean of the *k*-th observation, representing the shift of the ensemble mean induced by this observation,  $\Delta y'_{k,i}$  is the updated ensemble spread of the *k*-th observation, representing the reshaping of the model ensemble. They are respectively computed by

$$\bar{y}_{k}^{u} = \frac{(\sigma^{o})^{2}}{(\sigma^{o})^{2} + (\sigma_{k}^{p})^{2}} \bar{y}_{k}^{p} + \frac{(\sigma_{k}^{p})^{2}}{(\sigma^{o})^{2} + (\sigma_{k}^{p})^{2}} \bar{y}_{k}^{o} , \text{ and}$$

$$\Delta y_{k,i}^{\prime} = \sqrt{\frac{(\sigma^{o})^{2}}{(\sigma^{o})^{2} + (\sigma_{k}^{p})^{2}}} \left( y_{k,i}^{p} - \bar{y}_{k}^{p} \right) ,$$

$$(2)$$

where the first equation shows that the ensemble mean shifts toward the prior model ensemble mean  $\bar{y}_k^p$  or the observation value  $y_k^o$ , and whether it is  $\bar{y}_k^p$  or  $y_k^o$  depends on who has the smaller variance. The second equation denotes that the prior probability density function is squashed by a new observation.

The second step is to distribute the observational increments  $\Delta y_{k,i}^o$  on to the related model states x (a matrix of size  $N_{ens} \times N_{state}$ ) and this assimilation process can be expressed as

$$\Delta x_{j,i} = \frac{\cos(x_j, y_k^p)}{(\sigma_k^p)^2} \Delta y_{k,i}^o \quad , \tag{3}$$

where  $\Delta x_{i,j}$  is the contribution of the k-th observation to the *i*-th ensemble member of the *j*-th model variable  $x_{j,i}$  ( $j = 1 \sim N_{state}$ ).  $cov(x_j, y_k^p)$  is the error covariance between the prior ensemble of the *j*-th model variable  $x_j$  (in  $N_{ens}$  dimensions) and the prior (model-estimated) ensemble of the k-th observation  $y_k^p$  (in  $N_{ens}$  dimensions), and is calculated as

$$cov(x_j, y_k^p) = \frac{\sum_{i=1}^{Nens} (x_{j,i} - \bar{x}_j) (y_{k,i}^p - \bar{y}_k^p)}{N_{ens}}$$
, where  $\bar{x}_j$  is the ensemble mean of *j*-th model variable.

**(2) What dynamical model was used to propagate parameters in time? Was it identity model?**

RE: Thanks for this comment. There is no dynamical model used to propagate parameters in this paper and it is identity model. Addressing this comment, we have added more description and discussion about the parameter estimation in L96-97 and L221-231.

**L96-97:**

Observation-constrained model parameters no longer keep fixed values but are constantly varying over time.

**L221-231:**

The model parameters are fixed when parameter estimation is not performed. The parameters vary with observational information by parameter estimation. The core of the parameter estimation is to obtain the increment of the estimated parameter by a linear regression that is based on the error covariance between the prior parameter ensemble and the state ensemble (Anderson, 2001, 2003). The error covariance used in regression is flow dependent and temporally varying (Zhang and Anderson, 2003). Therefore, for the model parameter estimation, the observational increments are distributed onto a relevant parameter and the equation is

$$\Delta\beta_{j,i} = \frac{\cos(\beta_j, y_k^p)}{(\sigma_k^p)^2} \Delta y_{k,i}^o \quad , \tag{4}$$

where  $\Delta\beta_{j,i}$  is the contribution of the *k*-th observation to the *i*-th ensemble member of the *j*-th parameter being estimated, called  $\beta_{j,i}$   $(j = 1 \sim N_{para})$ .  $cov(\beta_j, y_k^p)$  is the error covariance between the prior ensemble of the *j*-th model parameter  $\beta_j$ (in  $N_{ens}$  dimensions) and the prior (model-estimated) ensemble of the *k*-th observation  $y_k^p$  (in  $N_{ens}$  dimensions), and is

calculated as  $cov(\beta_j, y_k^p) = \frac{\sum_{i=1}^{N_{ens}} (\beta_{j,i} - \bar{\beta}_j) (y_{k,i}^p - \bar{y}_k^p)}{N_{ens}}$ , where  $\bar{\beta}_j$  is the ensemble mean of *j*-th model parameter being optimized.

**Is there a more suitable model than identity? Please provide a brief reference review about different models used so far in literature and justify your choice.**

RE: Good suggestion! We have added a brief review and discussion about different models in L715-722.

**L715-722:**

Aksoy et al. (2006b) proposed a spatial updating technique that recovers the globally uniform parameter value using a spatial average of the entire spatially varying parameter field. Wu et al. (2012, 2013) explored the impact of the geographic dependence of observing system on the parameters. The adjustment of the parameters is based on the spatial distribution of the model state sensitivity to parameters. Liu et al. (2014a, b) proposed the adaptive spatial average method that obtains the final global uniform posterior parameter based on spatially varying posterior estimated parameter values. In this study, considering that the simple box models are used as a first step to explore AMOC transitions, it is more appropriate to use the identity model. The impact of geographic-dependent parameter optimization on climate estimation and prediction can be considered in future studies for complex systems such as CGCMs.

**(3) Have you applied any covariance inflation? Please explain.**

RE: Thanks for this suggestion! In this revision, a more detailed description of the covariance inflation scheme has been added in L239-246.

**L239-246:**

To further improve the signal-to-noise ratio of parameter estimation, Zhang (2011a) introduced an inflation scheme based on model sensitivity with respect to the parameter being estimated. In this inflation scheme, the inflation amplitude of a parameter ensemble is inversely proportional to the sensitivity. It is formulated as  $\tilde{\beta}_{j,i} = \bar{\beta}_j + max \left(1, \frac{\alpha_0 \sigma_0}{\sigma_j \sigma_t}\right) \left(\beta_{j,i} - \bar{\beta}_j\right)$ , where  $\tilde{\beta}_{j,i}$  denotes the inflated version of the *i*-th ensemble member of the *j*-th parameter being estimated,  $\sigma_0$  and  $\sigma_t$  are the prior ensemble spreads of this parameter at the initial time and time *t*,  $\alpha_0$  is a constant tuned by a trial-and-error procedure (e.g., Wu et al., 2016), and  $\sigma_j$  is the sensitivity of the model state with regard to *j*-th parameter. This indicates that if the prior ensemble spread of *j*-th parameter is smaller than  $\frac{\alpha_0}{\sigma_j}$  times the initial spread, it will be enlarged to this amount (e.g., Wu et al.,

2012; Han et al., 2014; Zhao et al., 2019).

**(4) Please address the issue of ensemble spread vs. forecast skill. Are they correlated in your experiments? Is ensemble spread over-estimated or under-estimated? Are ensembles collapsing?**

RE: Thanks for this comment. Ensemble spread and forecast skill are correlated in our experiments. There are two considerations regarding the ensemble spread of the parameter. As shown in the following figure, the ensemble spread is affected by the inflation level. On the one hand, overly small inflation factors and small ensemble spread will lead to too long fluctuation period of ensemble members, which affects the forecast skill and causes errors. Insufficient inflation may result in ensemble collapsing. On the other hand, overly large inflation factors lead the spread to jump out of the reasonable range. Considering the length limitation, the experiments on the ensemble spread and inflation factors are not shown in the manuscript. However, the relevant literature has been added in L244. Following this suggestion, Fig. 6 has been replaced by a new figure (Fig. S2d) in the revised manuscript. Accordingly, Figs. 5 and 7 have been updated due to the adjustment of the inflation scheme, and the new figures are almost identical to the original ones.

**L244:**

 $\alpha_0$  is a constant tuned by a trial-and-error procedure (e.g., Wu et al., 2016)

---

## Author Response (AR2)

**Response to Reviewer #1**

**Comments from Reviewer #1:**

**In this manuscript the authors describe a series of twin experiments in which the Ensemble Adjustment Kalman Filter (EAKF) is used for state and parameter estimation in models of the Atlantic Meridional Overturning Circulation (AMOC). The models in question are box models, driven by atmospheric input taken from the output of chaotic, rather than stochastic systems. The authors demonstrate that their methods allow reliable estimation of unknown parameters in the model AMOCs, which exhibit multiple stable equilibria. The results described here will be of interest to a broad segment of NPG readers, but the manuscript as it stands needs a great deal of work to make it acceptable.**

RE: A few conferences have been held for all co-authors to discuss the comments from reviewer #1. All authors converge to the point that the constructive comments and suggestions are extremely thoughtful and important for improving the manuscript and enhancing our understanding on the topic. Several extra experiments for addressing the concerns of the reviewer are performed. Thanks for your encouragement. All issues are replied point-by-point as below. We hope the whole manuscript has been essentially improved.

**Application of data assimilation methods to use observations to track the evolution of solutions of systems that exhibit multiple stable modes has been described by many authors since the 1990s; see, e.g., Weir et al., Nonlin Proc Geophys 2013 and references therein. A look through the literature since the 1990s will also turn up examples of simultaneous state and parameter estimation in simple systems. The novelty of the present work lies in the specific application to regime transitions in the AMOC. I don't know of other examples of application of optimized methods such as the EAKF to joint state and parameter estimation of box models of the AMOC, which have been around since Stommel (1961).**

RE: Excellent suggestion! We have added more detailed statements about the application of data assimilation methods (L111-118) and the examples of simultaneous state and parameter estimation in nonlinear systems having multiple stable modes (L122-128).

L111-118:
Tardif et al. (2014) implement data assimilation with EnKF to recover the AMOC with observations in a low-order coupled atmosphere-ocean climate model. They mainly explore the value of data assimilation for the initialization of the AMOC, while the effect of parameter errors in AMOC simulations needs further discussion. As another class of ensemble-based assimilation methods, particle filters, unlike the EnKF, are applicable to non-Gaussian probability distributions (e.g., Gordon et al., 1993; van Leeuwen, 2009). A mixture-based implicit particle method is presented and could detect transitions in an example with multiple attracting states (Weir et al., 2013a). However, the particle filter is plagued by the curse of dimensionality as the system dimension increases (Snyder et al., 2008; Carrassi et al., 2018).

L122-128:

Based on EAKF, a data assimilation scheme for enhanced parameter correction is designed to improve parameter estimation using observations (Zhang et al., 2012). Zhao et al. (2019) perform this scheme in a simple AMOC box model, and the model parameters are successfully optimized when the model errors are caused by only erroneously set parameters. Although the AMOC regime transition is not addressed in their study, their exploration of model sensitivities regarding parameters serves as a guideline for our research. Many efforts have been made to advance the application of data assimilation and parameter estimation in nonlinear systems having multiple equilibrium states (e.g., Miller et al., 1994, 1999; Khalil et al., 2009; Weir et al., 2013b; Bisaillon et al., 2015).

**The authors need to be more specific about exactly how their results differ from existing results, and why they are interesting. The closest thing that I can find to a statement of purpose appears at the end of the introduction, p4, beginning on line 111: "Here we present a method for improving the modeling of AMOC multi-equilibria. The new method is shown to simulate the AMOC transition between different equilibrium states accurately in two simple coupled models …" As noted above, others have shown the ability to simulate transition between different equilibrium states in other systems. If the authors' methods are novel, they should point out their differences from other methods that have been applied to similar systems.**

RE: Thank you for this suggestion! In this revision, we have reorganized the existing literature on AMOC transitions (L81-90), as well as more specifically illustrated how this study differs from previous studies and pointed out the novelty of this paper (L96-103, L128-133).

L81-90:

AMOC transitions can occur due to external forcing or internal feedback (Klockmann et al., 2020). The external forcing applied in systems may include freshwater forcing (e.g., Cessi, 1994; Castellana et al., 2019), wind forcing (e.g., Ashkenazy and Tziperman, 2007; Kleppin et al., 2015), ice sheet forcing (e.g., Zhang et al., 2014; Mitsui and Crucifix, 2017), $CO_2$ forcing (e.g., Zhang et al., 2017). The physical processes in the model are changed by external forcing, resulting in the transition between different states of the AMOC. For the AMOC model without external forcing, the transition is triggered by complex internal interactions within the model, such as salt oscillations (Peltier and Vettoretti, 2014), internal oceanic processes (Sévellec and Fedorov, 2014), thermohaline oscillations (Brown and Galbraith, 2016), intermittencies in the sea-ice cover (Gottwald, 2021). Regardless of whether it is due to external forcing or internal feedback, AMOC transitions could be influenced by complex physical processes in models, and the parameters involved in these physical processes are usually fixed.

L96-103:

Observation-constrained model parameters are no longer kept at fixed values but are constantly varying over time. The purpose of this paper is to explore whether the variations of observation-constrained parameters that allow the physical processes of model to evolve over time can bring the simulation results closer to the "observed" feature of regime transitions. The models in this paper are obtained by coupling AMOC box model with Lorenz's model, similar to the work by Roebber (1995) or

Gottwald (2021), where the variation of AMOC is driven by the chaotic dynamical system. The thermal mode and the reverse haline mode correspond to different equilibrium states of the AMOC. For simplicity, we will refer to these different states as the stronger AMOC (on-state) and the weaker AMOC (off-state) in simple conceptual models (e.g., Weijer et al., 2019).

L128-133:
Although numerical simulations of the AMOC eventually exhibit multiple equilibria, the AMOC is not an explicit model variable; rather, it is derived from model variables such as atmospheric wind, ocean temperature and salinity. Instead of adjusting AMOC directly, the model states are adjusted through data assimilation. When constraining model parameters by observational information, the parameters that constantly vary with observations may provide more diversity in the physical processes involved with AMOC regime transition, so that the model can simulate more AMOC transition paths.

**Their presentation of their three box model, equations (5)-(10) is confusing. Why different systems of equations for the thermal and saline modes? For details of the model they refer the reader to Shen et al. (2011) and the model described there is a single system that exhibits both saline and thermal modes. The model they finally use, defined in equation (12), is a complex system with many switches. I don't understand why this is necessary. Why not just use some form of (5) from Shen et al. (2011)?**

RE: Sorry we haven't explained this clearly. Shen et al. (2011) described two different systems of equations for the thermal and haline modes, which are exactly the same as the equations (5)-(10) in this paper. They do not show the equations for the haline mode, but describe the equations for the thermal mode in detail. The haline mode is described briefly in a sentence between equation (2d) and equation (3) in Shen et al. (2011). Following this suggestion, we have added new lines to clarify this issue in L301-303, L318-320, and L401-402.

L301-303:
The governing equations for the haline mode also follow the study of Shen et al. (2011). They didn't show those equations, but only described them briefly. In this paper, to describe the construction of MOC3B-5V more clearly later, those equations are shown here.

L318-320:
Similar equations for the thermal and haline modes could be found in Guan and Huang (2008) for Eq. (1) (thermal mode) and Eq. (2) (haline mode), and in Shen and Guan (2015) for Eqs. (1)-(6) (thermal mode) and Eqs. (7)-(9) (haline mode).

L401-402:
A similar AMOC box model with many switches could be found in Castellana et al., (2019).

**The results they get from this model, driven by a chaotic atmosphere, are encouraging. Figures 5-7 show that the data assimilation/parameter estimation system works well, reproducing the "true" trajectory quite accurately and producing a good estimate of the unknown parameter, while leaving out the parameter estimation steps, and using EAKF for state estimation without adjusting the parameter to its true value does not do nearly so well.**

RE: Thanks for your encouragement.

**At the end of this section the authors state: "The MOC3B-5V model is just a simple conceptual model, and the model states x2, w, and η simply conceptually simulate the variation characteristics of the atmosphere and the ocean. Although the transitions of AMOC are simulated by the MOC3B-5V model, the specific physical meaning of the model is not explicit enough. The method of capturing regime transitions in Sect. 2 is proved to be feasible in the simple model, and the next step is to apply the method to a physics-based MOC box model." The next section describes experiments with a "physics-based MOC box model," which is no more complicated than the MOC3B-5V model in the previous section, and the authors do not make clear what conceptual points are made with the MOC3B-5V model that are any less clear in the "MOCBM." It seems to me that the sections dealing with the MOC3B-5V model, i.e., much of section 2 and all of section 3 could be left out entirely without any loss of understanding on the part of the reader, however convincing figures 5-7 may be. It shouldn't be too hard for the authors to include additional figures corresponding to figures 5b, 6 and 7 to illustrate the details of the MOCBM experiment**

RE: Thanks for your thoughtful suggestion! In this revision, we address this comment by two parts. In the first part, we like to show the details of the MOCBM experiment (additional Figures S1, S2, and S3 below corresponding to Figures 5, 6 and 7 in the manuscript). Based on your suggestion, we have added Fig. S1b that shows the assimilation result to the revised manuscript (Fig. 9b) and more relevant discussions have been added in L614-616. Given that Figs. S2 and S3 are very similar to the behavior of the MOC3B-5V model, they are not shown in the manuscript. However, the corresponding descriptions have been added in L619-620.

In the second part, we have added more discussions to justify the necessity of MOC3B-5V and MOCBM models as three aspects. First, new lines have been added to emphasize the benefits of the MOC3B-5V model in L520-524. Second, more discussions about the important differences between these two models have been added in L621-624. Third, we have added more description and discussion about the different behaviors of different models in data assimilation and parameter estimation to optimize the simulation results in L665-676.

L614-616:
Figure 9b shows the time series of the MOC value in the "truth" simulation (red lines) and in the free model control ensemble simulations with only data assimilation. Due to the existence of parameter error, inaccurate analyses are obtained when only data assimilation was performed without parameter estimation.

L619-620:

Since the behavior of the MOCBM and MOC3B-5V models are very similar, the figures corresponding to Fig. 6 and Fig. 7 are not shown here.

L520-524:

The 5-variable conceptual climate model could simulate the interactions between the atmosphere and ocean, and coupling it with the three-box MOC model could accurately address the main questions in this paper. The transferring of the uncertainty of the MOC3B-5V model is particularly simple and easily understood. With the help of this model, we found that the coupled model parameter estimation with observations can significantly mitigate the model deviations, thus capturing regime transitions of the AMOC. As such, the main outcome of this paper can be more readily demonstrated with this simple model.

L621-624:

The box model in the MOCBM is based on the classical approach of adopting a buoyancy constraint, and the circulation is regulated by the surface buoyancy difference, implying that surface thermohaline forcing drives AMOC (Birchfield, 1989). In contrast, for the box model in the MOC3B-5V model, the constraint is based on mechanical energy sustaining diapycnal mixing, and the circulation is maintained by mechanical energy from wind stress and tides (Shen et al., 2011).

L665-676:

Since the circulation is driven only by the meridional gradients of the upper ocean temperature and salinity in the buoyancy-constraint MOCBM model, AMOC regime transitions can be captured to some extent when the upper ocean temperature and salinity are directly adjusted by data assimilation only, but the simulation results are not accurate enough. In this simple model, since the data assimilation has worked well, the contribution of parameter estimation is relatively small but still indispensable. The AMOC regime transitions are captured more accurately by parameter estimation. The degree of contributions of data assimilation or parameter estimation to the optimization of simulation results is different in these two models. Compared with the MOCBM model, the energy-constraint MOC3B-5V model is more representative for the role of parameter estimation because the circulation is maintained by mechanical energy. When leaving out the parameter estimation steps and constraining the model states only by data assimilation, the accuracy of state estimation is not high due to the existence of parameter errors. Given the fact that the circulation is driven in a more complex way in the real world, this simple model study only provides a conceptual understanding and guideline for more complex real systems such as Coupled General Circulation Model (CGCM).

[Figure]

**Figure S1**. Time series of the meridional overturning circulation $q$ in the "truth" simulation with $\gamma = 0.06364$ (red) and the individual ensemble members (orange) in the free model control ensemble simulations a) without data assimilation and parameter estimation, b) with data assimilation or c) with data assimilation and parameter estimation using erroneously-guessed $\gamma$ values with a Gaussian perturbation that has a mean value of 0.070004 and a standard deviation of 10% times the standard value. The dashed black line denoting $q = 0$ is a division line between two equilibrium states.

[Figure]

**Figure S2**. Time series of the estimated $\gamma$ values with a Gaussian perturbation that has a mean value of 0.070004 and a standard deviation of 10% times the standard value in the individual ensemble members (orange) in the free model control ensemble simulations with data assimilation and parameter estimation. The solid red line denoting $\gamma = 0.06364$ marks the true value of $\gamma$ being estimated. The dotted-dashed black line denoting 800th-year marks the start of parameter estimation using the observations of the atmosphere states and the temperature and salinity of the surface ocean.

[Figure]

**Figure S3**. Time series of the meridional overturning circulation $q$ in one of the ensemble members in the free model control ensemble simulations with (red) or without (orange) data assimilation and parameter estimation, where $\gamma$ is erroneously guessed as 0.067419. The dashed black line denoting $q = 0$ is a division line between the thermal mode equilibrium state and the haline mode equilibrium state.

**The two systems dealt with here are highly parameterized, but the parameter the authors chose to estimate, in both cases, was a parameter in the atmospheric model, equations (11) and (16). Why didn't they choose a parameter in the box models?**

RE: Thanks for your suggestion. We have re-chosen the parameter to be estimated in the second system. The parameter $b$ in the atmospheric model is replaced by the parameter $\gamma$ in the box model. Following this suggestion, the first paragraph in Section 4.2 has been rewritten (L596-599), and Fig. 9 has been replaced by a new figure (L604-609). The new results are similar to the original results for parameter $b$. Both can demonstrate that the AMOC regime transitions can be captured by constraining the model parameters with observations.

Besides, the discussion about parameter selection has been added in L599-603. Considering the length limitation, the process of investigating the model sensitivity is not shown in the manuscript. However, the relevant literature has been added in L602-603. The process of investigating the model sensitivity will be described later. The results are shown in Figure S4 and the most sensitive parameter is $\gamma$.

L596-599:
In the twin experimental framework, the assimilation model is similar to the truth model except that parameter γ in the box model is assumed to be incorrectly estimated, with error that is 10% greater than the standard value 0.06364. Thus, the mean value of all parameters from the twenty assimilation models is 0.070004 and their standard deviation is 10% of the standard value.

L604-609: (The same as Fig. S1).

L599-603:
The parameters in the atmosphere model or in the ocean model could be selected for parameter estimation to address the points in this paper. Given that we have experimented with the parameter in the atmosphere model before, here we show the experiment with the parameter of the ocean model. Again, the parameter being estimated is based on the model sensitivities regarding all parameters in the box model (Zhao et al., 2019).

L602-603:
Again, the parameter being estimated is based on the model sensitivities regarding all parameters in the box model (Zhao et al., 2019).

[Figure]

**Figure S4**. The sensitivity percentage of MOC in the space of the parameters. The sensitivity percentage refers to the ratio of the specified parameter sensitivity to the total sum of the sensitivities for the 11 parameters in the box model. The time-averaged value of the MOC spread is calculated by data over the last 200 years.

Referring to Section 3.2.3 in Zhao et al. (2019), the process of investigating the model sensitivity regarding each parameter is as follows. Each parameter is individually perturbed into 20 members from the ensemble as white noise, with a standard deviation of 10% of its standard value, while the remaining 10 parameters remain fixed at their standard values. Then, from the same initial states, all 20 ensemble members are freely run for 250 years. Only the model outputs of the last 200 years are used to quantitatively calculate the relative sensitivities (Owing to the MOC value is a long-time-scale variable). We use the standard deviation in the MOC strength to evaluate the model sensitivity regarding each parameter. The time-averaged sensitivity percentages of the MOC strength with respect to all parameters are shown in Fig. S4.

**References**

[revised manuscript text omitted]

**Response to Reviewer #2**

**Comments from Reviewer #2:**

**General Comments:**

**This paper addresses regime transition in AMOC though state and parameter estimation, via application of the EAKF. While the idea of parameter estimation is not new in ensemble-based data assimilation and there are many published papers addressing it, I believe that the novelty of the paper is the application of EAKF to regime transition in AMOC. Still, the paper needs to explain what the main findings are and how those findings could improve our knowledge about AMOC. In addition, I have some specific comments requiring clarification of the parameter estimation approach and the EAKF equations.**

RE: A few conferences of all co-authors have been held to discuss the comments of reviewer #2. All authors appreciate greatly for the encouragements and comments. All the comments are very important and useful for authors to improve the quality of this manuscript. The paper is renewed as the reviewer's suggestions. Thanks for your encouragement. In this revision, a more detailed explanation has been added to the Introduction section. We have expressed better what the main findings of this paper are (L96-103, L665-676) and how these findings could improve our knowledge about AMOC (L128-133). All specific comments are replied point-by-point as below. We hope the whole manuscript has been essentially improved.

L96-103:
Observation-constrained model parameters are no longer kept at fixed values but are constantly varying over time. The purpose of this paper is to explore whether the variations of observation-constrained parameters that allow the physical processes of model to evolve over time can bring the simulation results closer to the "observed" feature of regime transitions. The models in this paper are obtained by coupling AMOC box model with Lorenz's model, similar to the work by Roebber (1995) or Gottwald (2021), where the variation of AMOC is driven by the chaotic dynamical system. The thermal mode and the reverse haline mode correspond to different equilibrium states of the AMOC. For simplicity, we will refer to these different states as the stronger AMOC (on-state) and the weaker AMOC (off-state) in simple conceptual models (e.g., Weijer et al., 2019).

L665-676:
Since the circulation is driven only by the meridional gradients of the upper ocean temperature and salinity in the buoyancy-constraint MOCBM model, AMOC regime transitions can be captured to some extent when the upper ocean temperature and salinity are directly adjusted by data assimilation only, but the simulation results are not accurate enough. In this simple model, since the data assimilation has worked well, the contribution of parameter estimation is relatively small but still indispensable. The AMOC regime transitions are captured more accurately by parameter estimation. The degree of contributions of data assimilation or parameter estimation to the optimization of simulation results is different in these two models. Compared with

the MOCBM model, the energy-constraint MOC3B-5V model is more representative for the role of parameter estimation because the circulation is maintained by mechanical energy. When leaving out the parameter estimation steps and constraining the model states only by data assimilation, the accuracy of state estimation is not high due to the existence of parameter errors. Given the fact that the circulation is driven in a more complex way in the real world, this simple model study only provides a conceptual understanding and guideline for more complex real systems such as Coupled General Circulation Model (CGCM).

L128-133:
Although numerical simulations of the AMOC eventually exhibit multiple equilibria, the AMOC is not an explicit model variable; rather, it is derived from model variables such as atmospheric wind, ocean temperature and salinity. Instead of adjusting AMOC directly, the model states are adjusted through data assimilation. When constraining model parameters by observational information, the parameters that constantly vary with observations may provide more diversity in the physical processes involved with AMOC regime transition, so that the model can simulate more AMOC transition paths.

**Specific Comments:**

**(1) Please explain how equations (1)-(4) were derived from the EAKF.**

RE: Good suggestion! In this revision, a detailed derivation process is shown below. Given the length limitation, these derivations are not added to the manuscript. However, a concise description and the corresponding references have been added in L192-197.

L192-197:
The ensemble adjustment Kalman filter (Anderson, 2001) is used for data assimilation and parameter estimation in this study. The basic process of the two-step EAKF (Anderson, 2003; Zhang and Anderson, 2003; Zhang et al., 2007) is to project the observational increment onto model states (relevant parameters) by calculating the error covariance between the prior ensemble of the model variable (parameter) and the model-estimated ensemble. The core of the two-step EAKF is to calculate the increment of each state variable by a local least squares fit (linear regression), and the calculation of the observational increment is related to the scalar application of the equations of EAKF (Anderson, 2003).

Based on the EAKF (Anderson, 2001), Anderson (2003) described a two-step data assimilation procedure for ensemble filtering under a local least squares framework.

The joint state-observation space is defined by the joint space state vector: $\mathbf{z} = [\mathbf{x}, \mathbf{y}]$, where $\mathbf{x}$ is the model state vector; $\mathbf{y} = h(\mathbf{x})$, where $h$ is the forward observation operator. Using Bayesian statistics, the distribution of the posterior (or updated) distribution can be computed from the prior distribution, as
$$\mathbf{p}(\mathbf{z}^u) = \mathbf{p}(\mathbf{y}^o|\mathbf{z}^p)\mathbf{p}(\mathbf{z}^p)/(\text{norm}) \ . \tag{S1}$$
At the heart of the ensemble Kalman filter is the fact that the product of the joint prior Gaussian with mean $\mathbf{\bar{z}}^p$, covariance

$\Sigma^p$, and the Gaussian observation distribution with mean $\mathbf{y}^o$ and error variance $\mathbf{R}$ has covariance

$$\Sigma^u = [(\Sigma^p)^{-1} + \mathbf{H}^T\mathbf{R}^{-1}\mathbf{H}]^{-1} \; , \tag{S2}$$

and mean

$$\bar{\mathbf{z}}^u = \Sigma^u[(\Sigma^p)^{-1}\bar{\mathbf{z}}^p + \mathbf{H}^T\mathbf{R}^{-1}\mathbf{y}^o] \; . \tag{S3}$$

The EAKF constructs an updated ensemble with a mean and sample variance that satisfy Eq. (S2) and Eq. (S3). In Anderson (2001), this is done by shifting the mean of the ensemble and then adjusting the spread of the ensemble around the updated mean using a linear operator $\mathbf{A}$:

$$\mathbf{z}_i^u = \mathbf{A}\big(\mathbf{z}_i^p - \bar{\mathbf{z}}^p\big) + \bar{\mathbf{z}}^u \tag{S4}$$

where $\mathbf{A}$ satisfies $\Sigma^u = \mathbf{A}\Sigma^p\mathbf{A}^T$.

In everything that follows, results are presented only for assimilation of a single scalar observation. Define the joint state space forward observation operator for a single observation as the order $1 \times k$ linear operator $\mathbf{H} = [0,0,\ldots,0,1]$, where $k$ is the joint state space size. The updated probability for the marginal distribution of the observation joint state variable y can be formed: $p_y(y^u) = p(y^o|y^p)p_y(y^p)/(\text{norm})$, where the subscript on the probability densities indicates a marginal probability on the observation variable. Note that this equation does not depend on any of the model state variables. This suggests a partitioning of the assimilation of an observation into two parts. The first determines updated ensemble members for the observation variable given the observation. To update the ensemble sample of $y^p$, an increment is computed for each ensemble member:

$$\Delta y_i = y_i^u - y_i^p \; , \tag{S5}$$

where $i = 1, \ldots, N$, and $N$ is the ensemble size. The second step computes corresponding increments for $i$-th ensemble sample of $j$-th state variable $\Delta x_{i,j}$. This requires assumptions that the prior distribution is Gaussian. This equivalent to assuming that a least squares fit to the prior ensemble members summarizes the relationship between the joint state variables.

[Figure]

**Figure S1.** An idealized representation showing the relation between update increments for a state variable $x$ and an observation variable $y$ for a five member ensemble represented by asterisks. The projection of the ensemble on the $x$ and $y$ axes is represented by a plus sign and the observation $y^o$ is represented by "$*$". The gray dashed line shows a global least squares fit to the ensemble members. Update increments for ensemble members 1 and 4 for $y$ are shown along with corresponding increments for the ensemble as a whole (thin vectors parallel to least squares fit) and for the $x$ ensemble. From Anderson (2003)

Figure S1 depicts the simplest example in which there is only a single state variable $x$. The observation variable $y$ is related to $x$ by the operator $h$, which is nonlinear in the figure. Increments for each ensemble sample of $y$ have been computed. The corresponding increments for $x$ are then computed by a least squares fit (linear regression) so that

$$\Delta x_i = \frac{cov(x_i, y_i)}{(\sigma^p)^2} \Delta y_i \ , \tag{S6}$$

where $cov(x_i, y_i)$ is the prior covariance of $x$ with $y$, $(\sigma^p)^2$ is the prior variance of $y$.

Equation (S1) implies that the observation variable can be updated independently of the other joint state variables. Using a scalar application of Eq. (S2), the updated variance for $y$ can be written

$$(\sigma^u)^2 = [((\sigma^p)^2)^{-1} + ((\sigma^o)^2)^{-1}]^{-1} = \frac{(\sigma^o)^2(\sigma^p)^2}{(\sigma^o)^2 + (\sigma^p)^2} \ . \tag{S7}$$

Applying a scalar version of Eq. (S3) to compute the updated mean

$$\bar{y}^u = (\sigma^u)^2[((\sigma^p)^2)^{-1}\bar{y}^p + ((\sigma^o)^2)^{-1}y^o]$$

$$= \frac{(\sigma^o)^2}{(\sigma^o)^2 + (\sigma^p)^2}\bar{y}^p + \frac{(\sigma^p)^2}{(\sigma^o)^2 + (\sigma^p)^2}y^o \ . \tag{S8}$$

Using a scalar application of Eq. (S4), the updated value of y can be written

$$y_i^u = \alpha(y_i^p - \bar{y}^p) + \bar{y}^u \ , \tag{S9}$$

where $\alpha = \sqrt{(\sigma^u)^2/(\sigma^p)^2} = \sqrt{\frac{(\sigma^o)^2}{(\sigma^o)^2 + (\sigma^p)^2}} \ .$

Hence, the observational increment from Eq. (S5) can be written

$$\Delta y_i = y_i^u - y_i^p = \alpha(y_i^p - \bar{y}^p) + \bar{y}^u - y_i^p$$

$$= \sqrt{\frac{(\sigma^o)^2}{(\sigma^o)^2 + (\sigma^p)^2}}(y_i^p - \bar{y}^p) + \frac{(\sigma^o)^2}{(\sigma^o)^2 + (\sigma^p)^2}\bar{y}^p + \frac{(\sigma^p)^2}{(\sigma^o)^2 + (\sigma^p)^2}y^o - y_i^p \ . \tag{S10}$$

For more details, please see Sect. 2c and 3b in Anderson (2003).

**Also, please explain exact meaning of each variable. For example, is $\Delta\beta^p$ a value of the parameter $\beta$, or prior ensemble perturbation of the parameter $\beta$? Is $\Delta y_i^p$ prior ensemble spread or prior ensemble perturbation? In addition, all vectors and matrices need to have clearly defined space (e.g., observation or state space) and dimensions (e.g., Nobs, Nstate, Nens, Nobs x Nens, ...).**

RE: Excellent suggestion! To fully address this comment, we have substantially rewritten Section 2.2. The meaning of each variable has been explained more exactly, and the space and dimensions of all vectors and matrices have been clearly defined. We believe that the new Section 2.2 has been considerably improved (L197-220).

L197-220:

All observations at time $t$ have the observation value $\boldsymbol{y^o}$ (in $N_{obs}$ dimensions). For a single observation $y_k^o$ at the $k$-th observation location ($k = 1 \sim N_{obs}$), the standard deviation of observational error is $\sigma^o$ (assumed to be Gaussian). The model states are mapped onto the observational space by applying a linear interpolation, and then the prior (model-estimated) ensemble of the $k$-th observation $\boldsymbol{y_k^p}$ (in $N_{ens}$ dimensions) can be obtained. $y_{k,i}^p$ is the $i$-th prior ensemble member of the $k$-th observation. The ensemble mean and standard deviation are $\bar{y}_k^p$ and $\sigma_k^p$, respectively.

The first step is to compute the observational increment of the $k$-th observation ($k = 1 \sim N_{obs}$). The observational increment $\Delta y_{k,i}^o$ for the $i$-th ensemble member ($i = 1 \sim N_{ens}$) is formulated by

$$\Delta y_{k,i}^o = \bar{y}_k^u + \Delta y_{k,i}' - y_{k,i}^p \ , \tag{1}$$

where $\bar{y}_k^u$ is the posterior ensemble mean of the $k$-th observation, representing the shift of the ensemble mean induced by this observation, $\Delta y_{k,i}'$ is the updated ensemble spread of the $k$-th observation, representing the reshaping of the model ensemble. They are respectively computed by

$$\bar{y}_k^u = \frac{(\sigma^o)^2}{(\sigma^o)^2 + (\sigma_k^p)^2} \bar{y}_k^p + \frac{(\sigma_k^p)^2}{(\sigma^o)^2 + (\sigma_k^p)^2} y_k^o \ , \text{ and}$$

$$\Delta y_{k,i}' = \sqrt{\frac{(\sigma^o)^2}{(\sigma^o)^2 + (\sigma_k^p)^2}} \left( y_{k,i}^p - \bar{y}_k^p \right) \ , \tag{2}$$

where the first equation shows whether the ensemble mean shifts closer to the prior model ensemble mean $\bar{y}_k^p$ or the observation value $y_k^o$, and whether it is $\bar{y}_k^p$ or $y_k^o$ depends on which has the smaller variance. The second equation denotes that the prior probability density function is squashed by a new observation.

The second step is to distribute the observational increments $\Delta y_{k,i}^o$ on to the related model states $\boldsymbol{x}$ (a matrix of size $N_{ens} \times N_{state}$) and this assimilation process can be expressed as

$$\Delta x_{j,i} = \frac{cov(x_j, y_k^p)}{(\sigma_k^p)^2} \Delta y_{k,i}^o \ , \tag{3}$$

where $\Delta x_{j,i}$ is the contribution of the $k$-th observation to the $i$-th ensemble member of the $j$-th model variable $x_{j,i}$ ($j = 1 \sim N_{state}$). $cov(\boldsymbol{x_j}, \boldsymbol{y_k^p})$ is the error covariance between the prior ensemble of the $j$-th model variable $\boldsymbol{x_j}$ (in $N_{ens}$ dimensions) and the prior (model-estimated) ensemble of the $k$-th observation $\boldsymbol{y_k^p}$ (in $N_{ens}$ dimensions), and is calculated as

$$cov(\boldsymbol{x_j}, \boldsymbol{y_k^p}) = \frac{\sum_{i=1}^{N_{ens}} (x_{j,i} - \bar{x}_j)(y_{k,i}^p - \bar{y}_k^p)}{N_{ens}} \ , \text{ where } \bar{x}_j \text{ is the ensemble mean of } j\text{-th model variable.}$$

**(2) What dynamical model was used to propagate parameters in time? Was it identity model?**

RE: Thanks for this comment. There is no dynamical model used to propagate parameters in this paper and it is identity model. Addressing this comment, we have added more description and discussion about the parameter estimation in L96 and L221-231.

L96:
Observation-constrained model parameters are no longer kept at fixed values but are constantly varying over time.

L221-231:

The model parameters are fixed when parameter estimation is not performed. The parameters vary with observational information by parameter estimation. The core of the parameter estimation is to obtain the increment of the estimated parameter by a linear regression that is based on the error covariance between the prior parameter ensemble and the state ensemble (Anderson, 2001, 2003). The error covariance used in regression is flow dependent and temporally varying (Zhang and Anderson, 2003). Therefore, for the model parameter estimation, the observational increments are distributed onto a relevant parameter and the equation is

$$\Delta\beta_{j,i} = \frac{cov(\boldsymbol{\beta}_j, y_k^p)}{(\sigma_k^p)^2} \Delta y_{k,i}^o \ , \qquad (4)$$

where $\Delta\beta_{j,i}$ is the contribution of the $k$-th observation to the $i$-th ensemble member of the $j$-th parameter being estimated, called $\beta_{j,i}$ ($j = 1 \sim N_{para}$). $cov(\boldsymbol{\beta}_j, y_k^p)$ is the error covariance between the prior ensemble of the $j$-th model parameter $\boldsymbol{\beta}_j$ (in $N_{ens}$ dimensions) and the prior (model-estimated) ensemble of the $k$-th observation $y_k^p$ (in $N_{ens}$ dimensions), and is calculated as $cov(\boldsymbol{\beta}_j, y_k^p) = \frac{\sum_{i=1}^{N_{ens}}(\beta_{j,i}-\bar{\beta}_j)(y_{k,i}^p-\bar{y}_k^p)}{N_{ens}}$, where $\bar{\beta}_j$ is the ensemble mean of $j$-th model parameter being optimized.

**Is there a more suitable model than identity? Please provide a brief reference review about different models used so far in literature and justify your choice.**

RE: Good suggestion! We have added a brief review and discussion about different models in L716-723.

L716-723:

Aksoy et al. (2006b) proposed a spatial updating technique that recovers the globally uniform parameter value using a spatial average of the entire spatially varying parameter field. Wu et al. (2012, 2013) explored the impact of the geographic dependence of observing system on the parameters. The adjustment of the parameters is based on the spatial distribution of the model state sensitivity to parameters. Liu et al. (2014a, b) proposed the adaptive spatial average method that obtains the final global uniform posterior parameter based on spatially varying posterior estimated parameter values. In this study, considering that the simple box models are used as a first step to explore AMOC transitions, it is more appropriate to use the identity model. The impact of geographic-dependent parameter optimization on climate estimation and prediction can be considered in future studies for complex systems such as CGCMs.

**(3) Have you applied any covariance inflation? Please explain.**

RE: Thanks for this suggestion! In this revision, a more detailed description of the covariance inflation scheme has been added in L239-246.

L239-246:

To further improve the signal-to-noise ratio of parameter estimation, Zhang (2011a) introduced an inflation scheme based on model sensitivity with respect to the parameter being estimated. In this inflation scheme, the inflation amplitude of a parameter ensemble is inversely proportional to the sensitivity. It is formulated as $\tilde{\beta}_{j,i} = \bar{\beta}_j + max\left(1, \frac{\alpha_0 \sigma_0}{\sigma_j \sigma_t}\right)(\beta_{j,i} - \bar{\beta}_j)$ , where $\tilde{\beta}_{j,i}$ denotes the inflated version of the $i$-th ensemble member of the $j$-th parameter being estimated, $\sigma_0$ and $\sigma_t$ are the prior ensemble spreads of this parameter at the initial time and time $t$, $\alpha_0$ is a constant tuned by a trial-and-error procedure (e.g., Wu et al., 2016), and $\sigma_j$ is the sensitivity of the model state with regard to $j$-th parameter. This indicates that if the prior ensemble spread of $j$-th parameter is smaller than $\frac{\alpha_0}{\sigma_j}$ times the initial spread, it will be enlarged to this amount (e.g., Wu et al., 2012; Han et al., 2014; Zhao et al., 2019).

**(4) Please address the issue of ensemble spread vs. forecast skill. Are they correlated in your experiments? Is ensemble spread over-estimated or under-estimated? Are ensembles collapsing?**

RE: Thanks for this comment. Ensemble spread and forecast skill are correlated in our experiments. There are two considerations regarding the ensemble spread of the parameter. As shown in the following figure, the ensemble spread is affected by the inflation level. On the one hand, overly small inflation factors and small ensemble spread will lead to too long fluctuation period of ensemble members, which affects the forecast skill and causes errors. Insufficient inflation may result in ensemble collapsing. On the other hand, overly large inflation factors lead the spread to jump out of the reasonable range. Considering the length limitation, the experiments on the ensemble spread and inflation factors are not shown in the manuscript. However, the relevant literature has been added in L243-244. Following this suggestion, Fig. 6 has been replaced by a new figure (Fig. S2d) in the revised manuscript. Accordingly, Figs. 5 and 7 have been updated due to the adjustment of the inflation scheme, and the new figures are almost identical to the original ones.

L243-244:

[revised manuscript text omitted]

**Response to Editorial Comments**

**Editorial Comments:**

**L122-128:**
**Based on EAKF, a data assimilation scheme for enhancive parameter correction**
**Replace "enhancive" with "enhanced"**

RE: Thanks. Please see L122-123.
L122-123: "a data assimilation scheme for enhanced parameter correction".

**L520-524:**
**The 5-variable conceptual climate model could simulate the interactions between the atmosphere and ocean, and coupling it with the three-box MOC model could accurately address the ideas in this paper.**
**Replace "ideas" with "main questions"?**

RE: Thanks. Please see L521.
L521: "address the main questions in this paper".

**L520-524:**
**As such, this simple model has more visibility to demonstrate the essence of the problem.**
**"As such, the main outcome of this paper can be more readily demonstrated with this simple model."**

RE: Thanks. Please see L524.
L524: "As such, the main outcome of this paper can be more readily demonstrated with this simple model".

**L96-103:**
**Observation-constrained model parameters no longer keep fixed values but are constantly varying over time.**
**"Observation-constrained model parameters are no longer kept at fixed values…"**

RE: Thanks. Please see L96.
L96: "Observation-constrained model parameters are no longer kept at fixed values".

**L96-103:**
**can make the simulation results closer to the "observed" feature of regime transitions.**
**"can bring…"**

RE: Thanks. Please see L98.

L98: "can bring the simulation results closer to the "observed" feature of regime transitions".

**L128-133:**

**Although the AMOC model eventually exhibits multiple equilibria, the AMOC is not a direct model state...**

**Alternative text:**

**Although numerical simulations of the AMOC eventually exhibit multiple equilibria, the AMOC is not an explicit model variable; rather, it is derived from model variables such as…"**

RE: Thanks. Please see L128-130.

L128-130: "Although numerical simulations of the AMOC eventually exhibit multiple equilibria, the AMOC is not an explicit model variable; rather, it is derived from model variables such as".

**l.197-220:**

**where the first equation shows that the ensemble mean shifts toward the prior model ensemble mean $yyk\bar{k}\,pp$ or the observation value $yykkoo$, and whether it is $y\bar{y}kkpp$ or $yykkoo$ depends on who has the smaller variance.**

**Alternative: "where the first equation shows whether the ensemble mean shifts closer to the prior model ensemble mean $yyk\bar{k}\,pp$ or the observation value $yykkoo$, and whether it is $y\bar{y}kkpp$ or $yykkoo$ depends on which has the smaller variance."**

RE: Thanks. Please see L211-212.

L211-212: "where the first equation shows whether the ensemble mean shifts closer to the prior model ensemble mean $\bar{y}_k^p$ or the observation value $y_k^o$, and whether it is $\bar{y}_k^p$ or $y_k^o$ depends on which has the smaller variance".

RE: Thanks for your careful checking. We have corrected the typos and grammar errors you mentioned above and examined the manuscript word for word to make sure that there are no textual errors.

---

## Author Response (AR4)

**Response to Reviewer #1**

**Comments from Reviewer #1:**

**I must apologize to the authors and to the editors for the lateness of this review. It was clear from the authors' response to my first review that they interpreted my comments in the constructive spirit in which they were meant, and conscientiously responded. I was therefore confident that I would recommend publication of the revised version. However, I must ask for one further minor revision. My problem is that I don't understand the workings of the 3 box MOC model.**

**As the authors point out correctly, I did not fully understand the model of Shen et al. (2011), in that I did not see that a full AMOC model based on the work of Shen et al. would require switches. I now believe I see the necessity for switches, but I still don't understand the qualitative behavior of the 3 box AMOC model.**

**In particular, lines 315-320 need more explanation. Line 316:**
**"By solving the non-dimensional differential equations and the continuity equations, there are two stable solutions and one unstable solution, which means that the three-box model has three equilibrium states." I find this confusing. When you say "…the three box model has three equilibrium states," do you mean that (5) and (8), coupled with the continuity equation and the energy constraint, each has 3 solutions, two stable and one unstable? It seems more likely, from figure 3 that the thermal mode is stable within the parameter range under consideration, while the haline mode has a stable equilibrium state with near-zero overturning, and a stability transition on some set of critical points in parameter space. I might conclude from figure 3 that the haline and thermal modes have nearly equal overturning rates for e and omega greater than their critical values, and rho less than its critical value. Is this, in fact, the case?**

**Given the setup of the 3 box MOC model it is not clear what you mean by "The three box model has three equilibrium states," with 2 that are presumably stable and one that is not. I presume further that your model tracks the evolution of the overturning velocity by the energy constraint (14), and the system undergoes a transition when the overturning velocity changes sign.**

**Do I have this right? You really have to make this clear.**

RE: Thanks for your thoughtful comment! In this revision, we have made a clear and concise statement about the solutions of the three-box model and the sentence you mentioned has been revised (L315-321). Based on your suggestions, we have added new lines to clarify the question of the unstable solution in L357-364.

L315-321:

The time tendency in Eqs. (5) and (8) are set to be zero, and then the governing equations for the thermal mode or the haline mode are solved, respectively. Equations (5)-(7) have one stable solution, and Equations (8)-(10) have one stable solution and one unstable solution. Hence, the three-box model has a total of three mathematical solutions. This result obtained by solving the equations could be found in Shen et al. (2011).

L357-364:

The first aim of this study is to simulate the AMOC transition between different equilibrium states in the time series. However, a time series of overturning rate cannot be obtained by solving the governing equations after setting the time tendency in Eqs. (5) and (8) to zero. Therefore, without setting the time tendency to zero, we use a leapfrog time differencing scheme to forward the temperature and salinity to obtain the time series. For an unstable solution obtained by setting the time tendency to zero, a small perturbation on the solution will grow exponentially (Shen et al., 2011), so it cannot be obtained by using the time differencing scheme. Thus, the equilibrium states resolved through integrating time tendency equations in this study do not include the unstable solution described by Shen et al. (2011).

**References**

Shen, Y., Guan, Y. P., Liang, C. J., and Chen, D. K.: A three-box model of thermohaline circulation under the energy constraint, Chinese Phys. Lett., 28, 059201, https://doi.org/10.1088/0256-307x/28/5/059201, 2011.

**Response to Editorial Comments**

**Editorial Comments:**

**"using a leapfrog time differencing scheme to forward the temperature and salinity so as the overturning rate". - It appears a verb is missing from "so as the overturning rate"?**

RE: Thanks for your careful checking. We have revised this sentence in L360-361.

L360-361:
we use a leapfrog time differencing scheme to forward the temperature and salinity to obtain the time series.

**"a small perturbation on the solution will grow exponentially (Shen et al., 2011), so it is rather than a physical solution" - not clear what you mean by " so it is rather than a physical solution", please clarify.**

RE: Following your suggestion, the sentences have been rewritten (L361-364). Thanks.

L361-364:
For an unstable solution obtained by setting the time tendency to zero, a small perturbation on the solution will grow exponentially (Shen et al., 2011), so it cannot be obtained by using the time differencing scheme. Thus, the equilibrium states resolved through integrating time tendency equations in this study do not include the unstable solution described by Shen et al. (2011).